# Affordable AI Assistants with Knowledge Graph of Thoughts

## Abstract

Large Language Models (LLMs) are revolutionizing the development of AI assistants capable of performing diverse tasks across domains. However, current state-of-the-art LLM-driven agents face significant challenges, including high operational costs and limited success rates on complex benchmarks like GAIA. To address these issues, we propose Knowledge Graph of Thoughts (KGoT), an innovative AI assistant architecture that integrates LLM reasoning with dynamically constructed knowledge graphs (KGs). KGoT extracts and structures task-relevant knowledge into a dynamic KG representation, iteratively enhanced through external tools such as math solvers, web crawlers, and Python scripts. Such structured representation of task-relevant knowledge enables low-cost models to solve complex tasks effectively while also minimizing bias and noise. For example, KGoT achieves a 29% improvement in task success rates on the GAIA benchmark compared to Hugging Face Agents with GPT-4o mini. Moreover, harnessing a smaller model dramatically reduces operational costs by over $36\times$ compared to GPT-4o. Improvements for other models (e.g., Qwen2.5-32B and Deepseek-R1-70B) and benchmarks (e.g., SimpleQA) are similar. KGoT offers a scalable, affordable, versatile, and high-performing solution for AI assistants.

## 1 Introduction

Large Language Models (LLMs) are transforming the world. However, training LLMs is expensive, time-consuming, and resource-intensive. In order to democratize the access to generative AI, the landscape of agent systems has massively evolved during the last two years (LangChain Inc., 2025a; Rush, 2023; Kim et al., 2024; Sumers et al., 2024; Hong et al., 2024; Guo et al., 2024; Edge et al., 2025; Besta et al., 2025c; Zhuge et al., 2024; Beurer-Kellner et al., 2024; Shinn et al., 2023; Kagaya et al., 2024; Zhao et al., 2024a; Stengel-Eskin et al., 2024; Wu et al., 2024). These schemes have been applied to numerous tasks in reasoning (Creswell et al., 2023; Bhattacharjya et al., 2024; Besta et al., 2025c), planning (Wang et al., 2023c; Prasad et al., 2024; Shen et al., 2023; Huang et al., 2023), software development (Tang et al., 2024), and many others (Xie et al., 2024; Li & Vasarhelyi, 2024; Schick et al., 2023; Beurer-Kellner et al., 2023).

Among the most impactful applications of LLM agents is the development of AI assistants capable of helping with a wide variety of tasks. These assistants promise to serve as versatile tools, enhancing productivity and decision-making across domains. From aiding researchers with complex problem-solving to managing day-to-day tasks for individuals, AI assistants are becoming an indispensable part of modern life. Developing such systems is highly relevant, but remains challenging, particularly in designing solutions that are both effective and economically viable.

The GAIA benchmark (Mialon et al., 2024) has become a key standard for evaluating LLM-based agent systems across diverse tasks, including web navigation, code execution, image reasoning, scientific QA, and multimodal challenges. Despite its introduction nearly two years ago, top-performing solutions still struggle with many tasks. Moreover, operational costs remain high: running all validation tasks with Hugging Face Agents (Roucher & Petrov, 2025) and GPT-4o costs ≈$200, so more affordable alternatives are necessary. Smaller models like GPT-4o mini significantly reduce expenses but suffer from steep drops in task success, making them insufficient. Open large models also pose challenges due to demanding infrastructure needs, while smaller open models, though cheaper to run, lack sufficient capabilities.

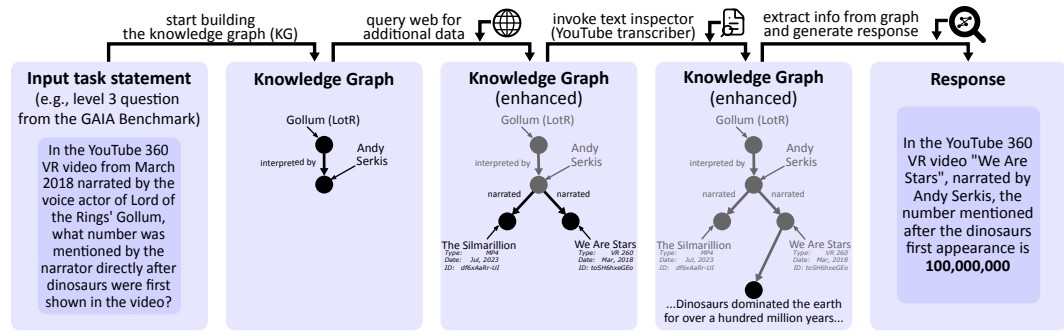

Figure 1: **The key idea behind Knowledge Graph of Thoughts (KGoT)**: transforming the representation of a task for an AI assistant from a textual form into a knowledge graph (KG). As an example, we use a Level-3 (i.e., highest difficulty) task from the GAIA benchmark. In order to solve the task, KGoT evolves this KG by adding relevant information that brings the task closer to completion. This is achieved by iteratively running various tools. Finally, the task is solved by extracting the relevant information from the KG, using – for example – a graph query, or an LLM's inference process with the KG provided as a part of the input prompt. More examples of KGs are in Appendix A.

To address these challenges, we propose Knowledge Graph of Thoughts (KGoT), a novel AI assistant architecture that significantly reduces task execution costs while maintaining a high success rate (**contribution #1**). The central innovation of KGoT lies in its use of a knowledge graph (KG) (Singhal, 2012; Besta et al., 2024b) to represent knowledge relevant to a given task. A KG organizes information into triples, providing a structured representation of knowledge that small, cost-effective models can efficiently process. Hence, KGoT "turns the unstructured into the structured", i.e., KGoT turns the often unstructured data such as website contents or PDF files into structured KG triples. This approach enhances the comprehension of task requirements, enabling even smaller models to achieve performance levels comparable to much larger counterparts, but at a fraction of the cost.

The KGoT architecture (**contribution #2**) implements this concept by iteratively constructing a KG from the task statement, incorporating tools as needed to gather relevant information. The constructed KG is kept in a graph store, serving as a repository of structured knowledge. Once sufficient information is gathered, the LLM attempts to solve the task by either directly embedding the KG in its context or querying the graph store for specific insights. This approach ensures that the LLM operates with a rich and structured knowledge base, improving its task-solving ability without incurring the high costs typically associated with large models. The architecture is modular and extensible towards different types of graph query languages and tools.

Our evaluation against top GAIA leaderboard baselines demonstrates its effectiveness and efficiency (**contribution #3**). KGoT with GPT-4o mini solves >2× more tasks from the validation set than Hugging Face Agents with GPT-4o or GPT-4o mini. Moreover, *harnessing a smaller model dramatically reduces operational costs: from $187 with GPT-4o to roughly $5 with GPT-4o mini*. KGoT's benefits generalize to other models, baselines, and benchmarks such as SimpleQA (Wei et al., 2024).

On top of that, KGoT reduces noise and simultaneously minimizes bias and improves fairness by externalizing reasoning into an explicit knowledge graph rather than relying solely on the LLM's internal generation (**contribution #4**). This ensures that key steps when resolving tasks are grounded in transparent, explainable, and auditable information.

## 2 KNOWLEDGE GRAPH OF THOUGHTS

We first illustrate the key idea, namely, using a knowledge graph to encode *structurally* the task contents. Figure 1 shows an example task and its corresponding evolving KG.

### 2.1 WHAT IS A KNOWLEDGE GRAPH?

A knowledge graph (KG) is a structured representation of information that organizes knowledge into a graph-based format, allowing for efficient querying, reasoning, and retrieval. Formally, a KG consists of a set of triples, where each triple $(s, p, o)$ represents a relationship between two entities $s$ (subject) and $o$ (object) through a predicate $p$. For example, the triple ("Earth", "orbits", "Sun") captures the fact that Earth orbits the Sun. Mathematically, a knowledge graph can be defined as a directed labeled graph $G = (V, E, L)$, where $V$ is the set of vertices (entities), $E \subseteq V \times V$ is the set of edges (relationships), and $L$ is the set of labels (predicates) assigned to the edges. Each entity or predicate may further include properties or attributes, enabling richer representation. Knowledge graphs are widely used in various domains, including search engines, recommendation systems, and AI reasoning, as they facilitate both efficient storage and complex queries.

## 2.2 Harnessing Knowledge Graphs for Effective AI Assistant Task Resolution

At the heart of KGoT is the process of transforming a task solution state into an evolving KG. The KG representation of the task is built from "thoughts" generated by the LLM. These "thoughts" are intermediate insights identified by the LLM as it works through the problem. Each thought contributes to expanding or refining the KG by adding vertices or edges that represent new information.

For example, consider the following Level 3 (i.e., highest difficulty) task from the GAIA benchmark: *"In the YouTube 360 VR video from March 2018 narrated by the voice actor of Lord of the Rings' Gollum, what number was mentioned by the narrator directly after dinosaurs were first shown in the video?"* (see Figure 1 for an overview; more examples of constructed KGs are in Appendix A). Here, the KG representation of the task solution state has a vertex *"Gollum (LotR)"*. Then, the thought *"Gollum from Lord of the Rings is interpreted by Andy Serkis"* results in adding a vertex for *"Andy Serkis"*, and linking *"Gollum (LotR)"* to *"Andy Serkis"* with the predicate *"interpreted by"*. Such integration of thought generation and KG construction creates a feedback loop where the KG continuously evolves as the task progresses, aligning the representation with problem requirements.

In order to evolve the KG task representation, KGoT iteratively interacts with tools and retrieves more information. For instance, the system might query the internet to identify videos narrated by Andy Serkis. It can also use a YouTube transcriber tool to find their publication date. This iterative refinement allows the KG to model the current "state" of a task at each step, creating a more complete and structured representation of this task.Once the KG has been sufficiently populated with task-specific knowledge, it serves as a robust resource for solving the problem.

## 2.3 Extracting Information from the KG

To accommodate different tasks, KGoT supports different ways to extract the information from the KG. Currently, we offer **graph query languages** or **general-purpose languages**; each of them can be combined with the so-called **Direct Retrieval**. First, one can use a graph query, prepared by the LLM in a language such as Cypher (Francis et al., 2018) or SPARQL (Pérez et al., 2009), to extract the answer to the task from the graph. This works particularly well for tasks that require retrieving specific patterns within the KG. Second, we also support general scripts prepared by the LLM in a general-purpose programming language such as Python. This approach, while not as effective as query languages for pattern matching, offers greater flexibility and may outperform the latter when a task requires, for example, traversing a long path in the graph. Third, in certain cases, once enough information is gathered into the KG, it may be more effective to directly paste the KG into the LLM context to solve the task. We refer to this approach as Direct Retrieval.

The above schemes offer a **tradeoff between accuracy, cost, and runtime**. For example, when low latency is priority, general-purpose languages should be used, as they provide an efficient lightweight representation of the KG and offer rapid access and modification of graph data. When token cost is most important, one should avoid Direct Retrieval (which consumes many tokens) and focus on either query or general-purpose languages, with a certain preference for the former, because its generated queries tend to be shorter than scripts. Finally, when aiming for solving as many tasks as possible, one should experiment with all three schemes. As shown in the Evaluation section, these methods have complementary strengths: Direct Retrieval is effective for broad contextual understanding, while graph queries and scripts are better suited for structured reasoning.

## 2.4 Bias, Fairness, and Noise Mitigation through KG-Based Representation

KGoT externalizes and structures the reasoning process, which reduces noise, mitigates model bias, and improves fairness, because in each iteration both the outputs from tools and LLM thoughts are converted into triples and stored explicitly. Unlike opaque monolithic LLM generations, this fosters transparency and facilitates identifying biased inference steps. It also facilitates noise mitigation: new triples can be explicitly checked for the quality of their information content before being integrated into the KG, and existing triples can also be removed if they are deemed redundant (examples of such triples that have been found and removed are in Appendix B.7).

## 3 System Architecture

The KGoT modular and flexible architecture, pictured in Figure 2, consists of three main components: the **Graph Store Module**, the **Controller**, and the **Integrated Tools**, each playing a critical role in the task-solving process. Below, we provide a detailed description of each component and its role in the system. Additional details are in Appendix B (architecture) and in Appendix C (prompts).

### 3.1 Maintaining the Knowledge Graph with the Graph Store Module

A key component of the KGoT system is the Graph Store Module, which manages the storage and retrieval of the dynamically evolving knowledge graph which represents the task state. In order to harness graph queries, we use a graph database backend; in the current KGoT implementation, we test Cypher together with Neo4j (Robinson et al., 2015), an established graph database (Besta et al., 2023a;b), as well as SPARQL together with the RDF4J backend (Ben Mahria et al., 2021). Then, in order to support graph accesses using a general-purpose language, KGoT harnesses the NetworkX library (NetworkX Developers, 2025) and Python. Note that the extensible design of KGoT enables seamless integration of any other backends and languages.

### 3.2 Managing the Workflow with the Controller Module

The Controller orchestrates the interactions between the KG and the tools. Upon receiving a user query, it iteratively interprets the task, determines the appropriate tools to invoke based on the KG state and task needs, and integrates tool outputs back into the KG. The Controller uses a dual-LLM architecture with a *clear separation of roles*: the **LLM Graph Executor** constructs and evolves the KG, while the **LLM Tool Executor** manages tool selection and execution.

The **LLM Graph Executor** determines the next steps after each iteration that constructs and evolves the KG. It identifies any missing information necessary to solve the task, formulates appropriate queries for the graph store interaction (retrieve/insert operations), and parses intermediate or final results for integration into the KG. It also prepares the final response to the user based on the KG.

The **LLM Tool Executor** operates as the executor of the plan devised by the LLM Graph Executor. It identifies the most suitable tools for retrieving missing information, considering factors such as tool availability, relevance, and the outcome of previous tool invocation attempts. For example, if a web crawler fails to retrieve certain data, the LLM Tool Executor might prioritize a different retrieval mechanism or adjust its queries. The LLM Tool Executor manages the tool execution process, including interacting with APIs, performing calculations, or extracting information, and returns the results to the LLM Graph Executor for further reasoning and integration into the KG.

### 3.3 Ensuring Versatile and Extensible Set of Integrated Tools

KGoT offers a hierarchical suite of tools tailored to diverse task needs. The **Python Code Tool** enables dynamic script generation and execution for complex computations. The **LLM Tool** supplements the controller's reasoning by integrating an auxiliary language model, enhancing knowledge access while minimizing hallucination risk. For multimodal inputs, the **Image Tool** supports image processing and extraction. Web-based tasks are handled by the **Surfer Agent** (based on the design by Hugging Face Agents (Roucher & Petrov, 2025)), which leverages tools like the **Wikipedia Tool**, **granular navigation tools** (PageUp, PageDown, Find), and SerpApi (SerpApi LLM, 2025) for search. Additional tools include the **ExtractZip Tool** for compressed files and the **Text Inspector Tool** for converting content from sources like MP3s and YouTube transcripts into Markdown. Finally, the user can seamlessly **add a new tool** by initializing the tool, passing in the logger object for tool use statistics, and appending the tool to the tool list via a Tool Manager object.

### 3.4 Ensuring High-Performance & Scalability

The used scalability optimizations include (1) **asynchronous execution** using asyncio (Python Software Foundation, 2025b) to parallelize LLM tool invocations, mitigating I/O bottlenecks and reducing idle time, (2) **graph operation parallelism** by reformulating LLM-generated Cypher queries to enable concurrent execution of independent operations in a graph database, and (3) MPI-based **distributed processing**, which decomposes workloads into atomic tasks distributed across ranks using a work-stealing algorithm to ensure balanced computational load and scalability.

### 3.5 Ensuring System Robustness

Robustness is ensured with two established mechanisms, **Self-Consistency** (Wang et al., 2023b) (via majority voting) and **LLM-as-a-Judge** (Gu et al., 2025) (other strategies such as embedding-based stability are also applicable (Besta et al., 2025d)). With Self-consistency, we query the LLM multiple times when deciding whether to insert more data into the KG or retrieve existing data, when deciding which tool to use, and when parsing the final solution. This approach reduces the impact of single-instance errors or inconsistencies in various parts of the KGoT architecture. LLM-as-a-Judge further reinforces the robustness, by directly employing the LLM agent to make these decisions based on generated reasoning chains.

Figure 2: **Architecture overview of KGoT (top part) and the design details combined with the workflow (bottom part).**

Overall, both Self-consistency and LLM-as-a-Judge have been shown to significantly enhance the robustness of prompting. For example, MT-Bench and Chatbot Arena show that strong judges (e.g., GPT-4 class) match human preferences at 80% agreement or more, on par with human-human agreement (Zheng et al., 2023). Prometheus/Prometheus-2 further demonstrate open evaluator LMs with the highest correlations to both humans and proprietary judges across direct-assessment and pairwise settings, and AlpacaEval has been validated against approximately 20K human annotations, addressing earlier concerns about reproducibility at scale. Similarly reliable gains have been shown for Self-consistency (Wang et al., 2023b).

## 4 SYSTEM WORKFLOW

We show the workflow in the bottom part of Figure 2, which begins when the user submits a problem to the system ❶. Next KGoT verifies whether the maximum number of iterations allowed for solving the problem has been reached ❷. If exceeded, the system will no longer try to gather additional information, but instead will return a solution with the existing data in the KG ❸. Otherwise, the majority vote (over several replies from the LLM) decides whether the system should proceed with the **Enhance** pathway (using tools to generate new knowledge) or directly proceed to the **Solve** pathway (gathering the existing knowledge in the KG and using it to deliver the task solution).

**The Enhance Pathway** If the majority vote indicates an Enhance pathway, the next step involves determining the tools necessary for completing the Enhance operation ❹. The system then orchestrates the appropriate tool calls based on the KG state ❺. Once the required data from the tools is collected, the system generates the Enhance query or queries to modify the KG appropriately. Each Enhance query is executed ❻ and its output is validated. If an error or invalid value is returned, the system attempts to fix the query using a decoder or the LLM, retrying a specified number of times. If retries fail, the query is discarded, and the operation moves on. After processing the Enhance operation, the system increments the iteration count and continues until the KG is sufficiently expanded or the iteration limit is reached. This path ensures that the knowledge graph is enriched with relevant and accurate information, enabling the system to progress toward a solution effectively.

**The Solve Pathway** If the majority vote directs the system to the Solve pathway, the system executes multiple solve operations iteratively ❼. If an execution produces an invalid value or error three times in a row, the system asks the LLM to attempt to correct the issue by recreating the used

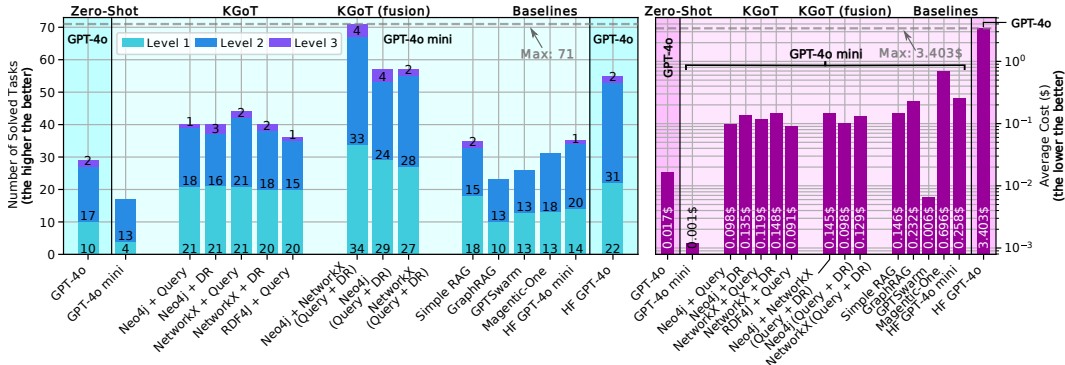

Figure 3: **Advantages of different variants of KGoT over other baselines** (Hugging Face Agents using both GPT-4o-mini and GPT-4o, Magentic-One, GPTSwarm, two RAG baselines, Zero-Shot GPT-4o mini, and Zero-Shot GPT-4o) on the validation dataset of the GAIA benchmark. DR stands for Direct Retrieval. The used model is GPT-4o mini unless noted otherwise.

query. The query is then re-executed. If errors persist after three such retries, the query is regenerated entirely, disregarding the faulty result, and the process restarts. After the Solve operation returns the result, final parsing is applied, which includes potential mathematical processing to resolve potential calculations **8** and refining the output (e.g., formatting the results appropriately) **9**.

## 5 EVALUATION

We now show advantages of KGoT over the state of the art. Additional results and full details on the evaluation setup are in Appendix D.

**Comparison Baselines.** We focus on the **Hugging Face (HF) Agents** (Roucher & Petrov, 2025), the most competitive scheme in the GAIA benchmark for the hardest level 3 tasks with the GPT-4 class of models. We also compare to two agentic frameworks, namely **GPTSwarm** (Zhuge et al., 2024) (a representative graph-enhanced multi-agent scheme) and **Magentic-One** (Fourney et al., 2024), an AI agent equipped with a central orchestrator and multiple integrated tool agents. Next, to evaluate whether database search outperforms graph-based knowledge extraction, we also consider two retrieval-augmented generation (RAG) (Lewis et al., 2020) schemes, a **simple RAG** scheme and **GraphRAG** (Edge et al., 2025). Both RAG baselines use the same tool-generated knowledge, chunking data at tool-call granularity (i.e., a chunk corresponds to individual tool call output). Simple RAG constructs a vector database from these tool outputs while GraphRAG instead models the tool outputs as a static KG of entities and relations, enabling retrieval via graph traversal. Finally, we use **Zero-Shot** schemes where a model answers without any additional agent framework.

**KGoT variants.** First, we experiment with **graph query languages** vs. **general-purpose languages**, cf. Section 2.3. For each option, we vary how the Solve operation is executed, by either having the LLM send a request to the backend (a Python script for NetworkX and a Cypher/SPARQL query for Neo4j/RDF4J) or by directly asking the LLM to infer the answer based on the KG (**Direct Retrieval (DR)**). We experiment with different query languages (**Cypher** vs. **SPARQL**). We also consider **"fusion" runs**, which simulate the effect from KGoT runs with both graph backends available simultaneously (or both Solve operation variants harnessed for each task). Fusion runs only incur negligible additional storage overhead because the generated KGs are small (up to several hundreds of nodes). Finally, we experiment with **different tool sets**. To focus on the differences coming from harnessing the KG, we reuse several utilities from AutoGen (Wu et al., 2024) such as Browser and MDConverter, and tools from HF Agents, such as Surfer Agent, web browsing tools, and Text Inspector.

**Considered Datasets** We use the GAIA benchmark (Mialon et al., 2024) focusing on the validation set (165 tasks) for budgetary reasons and also because it comes with the ground truth answers. The considered tasks are highly diverse in nature; many require parsing websites or analyzing PDF, image, and audio files. We focus on GAIA as this is currently the most comprehensive benchmark for general-purpose AI assistants, covering diverse domains such as web navigation, code execution, image reasoning, scientific QA, and multimodal tasks. We further evaluate on SimpleQA (Wei et al., 2024), a factuality benchmark of 4,326 questions, of which we sample 10% for budgetary reasons. The dataset spans diverse topics and emphasizes single, verifiable answers, making it effective for assessing factual accuracy.

## 5.1 Advantages of KGoT

Figure 3 shows the number of solved tasks (the left side) as well as the average cost per solved task (the right side) for different KGoT variants as well as all comparison baselines. While we focus on GPT-4o mini, we also show the results for HF Agents and Zero-Shot with GPT-4o. Additionally, we show the Pareto front in Figure 11 for the multidimensional optimization problem of improving accuracy (i.e., reducing failed tasks) and lowering cost. All variants of KGoT solve a greater number of tasks (up to 9 more) compared to HF Agents while also being more cost-efficient (between 42% to 62% lower costs). The key reason for the KGoT advantages stems from harnessing the knowledge graph–based representation of the evolving task state.

The ideal fusion runs of Neo4j and NetworkX solve an even greater number of tasks (57 for both) than the single runs, they have a lower average cost (up to 62% lower than HF Agents), and they even outperform HF Agents with GPT-4o. The fusion of all combinations of backend and solver types solve by far the highest number of tasks (71) – more than twice as much as HF Agents – while also exhibiting 44% lower cost than HF Agents. The direct Zero-Shot use of GPT-4o mini and GPT-4o has the lowest average cost per solved task (just $0.0013 and $0.0164 respectively), making it the most cost-effective, however this approach is only able to solve 17 and 29 tasks, respectively. GPTSwarm is cheaper compared to KGoT, but also comes with fewer solved tasks (only 26). While Magentic-One is a capable agent with a sophisticated architecture, its performance with GPT-4o mini is limited, solving 31 tasks correctly, while also exhibiting significantly higher costs. Simple RAG yields somewhat higher costs than KGoT and it solves fewer tasks (35). GraphRAG performs even worse, solving only 23 tasks and incurring even higher cost. While neither RAG baseline can invoke new tools to gather missing information (reducing accuracy and adaptability), GraphRAG's worse performance is due to the fact that it primarily targets query summarization and not tasks as diverse as those tested by GAIA. Overall, KGoT achieves the best cost-accuracy tradeoff, being both highly affordable and very effective.

## 5.2 Analysis of Methods for Knowledge Extraction

We explore different methods of extracting knowledge. Overall, in many situations, different methods have complementary strengths and weaknesses.

**Graph queries** with Neo4j excel at queries such as counting patterns. Yet, Cypher queries can be difficult to generate correctly, especially for graphs with more nodes and edges. Despite this, KGoT's Cypher queries are able to solve many new GAIA tasks that could not be solved without harnessing Cypher. SPARQL (Pérez et al., 2009) + RDF4J (Eclipse Foundation, 2025) is slightly worse (36 tasks solved) than Cypher + Neo4j (existing literature also indicates that LLMs have difficulties formulating effective SPARQL queries (Emonet et al., 2024; Mecharnia & d'Aquin, 2025)).

**Python** with NetworkX offers certain advantages over Neo4j by eliminating the need for a separate database server, making it a lightweight choice for the KG. Moreover, NetworkX computations are fast and efficient for small to medium-sized graphs without the overhead of database transactions. We observe that in cases where Neo4j-based implementations struggle, NetworkX-generated graphs tend to be more detailed and provide richer vertex properties and relationships. This is likely due to the greater flexibility of Python code over Cypher queries for graph insertion, enabling more fine-grained control over vertex attributes and relationships. Another reason may be the fact that Python is likely more represented in the training data of the respective models than Cypher.

Table 1: **Comparison of KGoT with other current state-of-the-art open-source agents on the full GAIA test set.** The baseline data, including for TapeAgent (Bahdanau et al., 2024), of the number of solved tasks is obtained through the GAIA Leaderboard (Mialon et al., 2025). We highlight the best performing scheme in given category in bold. Model: GPT-4o mini.

| **Agents** | **All** | **L1** | **L2** | **L3** |
|---|---|---|---|---|
| GPTSwarm | 33 | 15 | 15 | 3 |
| Magentic-One | 43 | 22 | 18 | 3 |
| TapeAgent | 66 | 28 | 35 | 3 |
| Hugging Face Agents | 68 | 30 | 34 | **4** |
| **KGoT (fusion)** | **73** | **33** | **36** | **4** |

Our analysis of failed tasks indicates that, in many cases, the KG contains the required data, but *the graph query fails to extract it*. Here, **Direct Retrieval**, where the entire KG is included in the model's context, performs significantly better by bypassing query composition issues. However, Direct Retrieval demonstrates lower accuracy in cases requiring structured, multi-step reasoning.

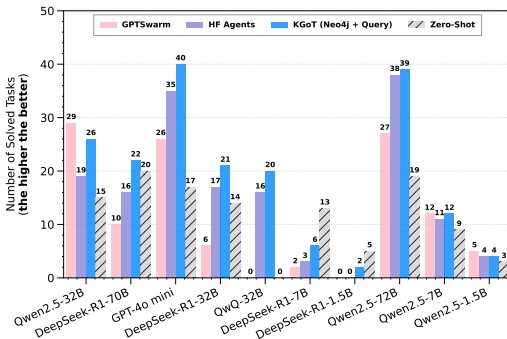
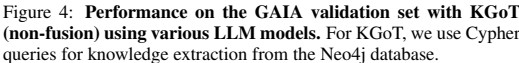

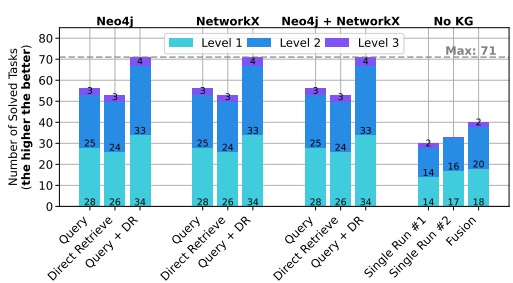

Figure 4: **Performance on the GAIA validation set with KGoT (non-fusion) using various LLM models.** For KGoT, we use Cypher queries for knowledge extraction from the Neo4j database.

Figure 5: **The impact coming from harnessing knowledge graphs (KGs) with different knowledge extraction methods (graph queries with Neo4j and Cypher, and general-purpose languages with Python and NetworkX), vs. using no KGs at all.** DR stands for Direct Retrieval. Model: GPT-4o mini.

### 5.3 ADVANTAGES ON THE GAIA TEST SET

Furthermore, our approach achieves state-of-the-art performance on the GAIA test set with the GPT-4o mini model. The results are shown in Table 1, underscoring its effectiveness across all evaluation levels. The test set consists of 301 tasks (93 Level 1 tasks, 159 Level 2 tasks and 49 Level tasks).

### 5.4 ADVANTAGES BEYOND GAIA BENCHMARK

We also evaluate KGoT as well as HF Agents and GPTSwarm on a 10% sample (433 tasks) of the SimpleQA benchmark (detailed results are in Appendix D.1). KGoT performs best, solving 73.21%, while HF Agents and GPTSwarm exhibit reduced accuracy (66.05% and 53.81% respectively). KGoT incurs only 0.018$ per solved task, less than a third of the HF Agents costs (0.058$), while being somewhat more expensive than GPTSwarm (0.00093$).

We further evaluate KGoT on the entire SimpleQA benchmark (due to very high costs of running all SimpleQA questions, we limit the full benchmark evaluation to KGoT). We observe no degradation in performance with a 70.34% accuracy rate. When compared against the official F1-scores of various OpenAI and Claude models (OpenAI, 2025), KGoT outperforms all the available results. Specifically, our design achieves a 71.06% F1 score, significantly surpassing the 49.4% outcome of the top-performing reasoning model and improving upon all mini-reasoning models by at least 3.5×. Furthermore, KGoT exceeds the performance of all standard OpenAI models, from GPT-4o's 40% F1 score to the best-scoring closed-source model, GPT-4.5, with 62.5%. More detailed results are available in Appendix D.1.

### 5.5 ENSURING SCALABILITY AND MITIGATING BOTTLENECKS

The primary bottleneck in KGoT arises from I/O-bound and latency-sensitive LLM tool invocations (e.g., web browsing, text parsing), which account for 72% of the runtime, which KGoT mitigates through asynchronous execution and graph operation parallelism as discussed in Section 3.4. A detailed breakdown of the runtime is reported in Appendix D.3. Figure 10 confirms KGoT's scalability, as increasing the number of parallelism consistently reduces the runtime. Moreover, due to the effective knowledge extraction process and the nature of the tasks considered, none of the tasks require large KGs. The maximum graph size that we observed was 522 nodes. This is orders of magnitude below any scalability concerns.

### 5.6 IMPACT FROM VARIOUS DESIGN DECISIONS

We also show the advantages of KGoT on **different open models** in Figure 4 over HF Agents and GPTSwarm for nearly all considered models (Yang et al., 2025; Guo et al., 2025). Interestingly, certain sizes of DeepSeek-R1 (Guo et al., 2025) offer high Zero-Shot performance that outperforms both KGoT and HF Agents, illustrating potential for further improvements specifically aimed at Reasoning Language Models (RLMs) (Besta et al., 2025a;c).

Finally, we investigate the **impact on performance coming from harnessing KGs, vs. using no KGs at all** (the "no KG" baseline), which we illustrate in Figure 5. Harnessing KGs has clear advantages, with a nearly 2× increase in the number of solved tasks. This confirms the positive impact from structuring the task related knowledge into a graph format, and implies that our workflow generates high quality graphs. To further confirm this, we additionally verified these graphs manually and we discovered that the generated KGs do contain the actual solution (e.g., the solution can be found across nodes/edges of a given KG by string matching). This illustrates that in the majority

of the solved tasks, the automatically generated KGs correctly represent the solution and directly enable solving a given task.

We offer further analyses in Appendix D, including studying the impact on performance from **different tool sets**, **prompt formats** as well as **fusion types**.

## 6 RELATED WORK

**Agent collaboration frameworks** are systems such as Magentic-One and numerous others (Zhuge et al., 2024; Tang et al., 2024; Liu et al., 2024; Li et al., 2024; Chu et al., 2024; Wu et al., 2024; Chen et al., 2024; Hong et al., 2024; Shinn et al., 2023; Zhu et al., 2024; Kagaya et al., 2024; Zhao et al., 2024a; Stengel-Eskin et al., 2024; Significant Gravitas, 2025; Zhu et al., 2025). The core KGoT idea that can be applied to enhance such frameworks is that a KG can also be used as a common shared task representation for multiple agents solving a task together. Such a graph would be then updated by more than a single agent. This idea proves effective, as confirmed by the fact that KGoT outperforms highly competitive baselines (HF Agents, Magentic-One, GPTSwarm) in both GAIA and SimpleQA benchmarks.

Some **agent frameworks explicitly use graphs** for more effective collaboration. Examples are GPTSwarm (Zhuge et al., 2024), MacNet (Qian et al., 2025), and AgentPrune (Zhang et al., 2025). These systems differ from KGoT as they use a graph to model and manage *multiple agents* in a structured way, forming a hierarchy of tools. Contrarily, KGoT uses KGs to represent *the task itself*, including its intermediate state. These two design choices are orthogonal and could be combined together. Moreover, while KGoT only relies on in-context learning; both MacNet (Qian et al., 2025) and AgentPrune (Zhang et al., 2025) require additional training rounds, making their integration and deployment more challenging and expensive than KGoT.

Many works exist in the domain of **general prompt engineering** (Beurer-Kellner et al., 2024; Besta et al., 2025c; Yao et al., 2023a; Besta et al., 2024a; Wei et al., 2022; Yao et al., 2023b; Chen et al., 2023; Creswell et al., 2023; Wang et al., 2023a; Hu et al., 2024; Dua et al., 2022; Jung et al., 2022; Ye et al., 2023). One could use such schemes to further enhance respective parts of the KGoT workflow. While we already use prompts that are suited for encoding knowledge graphs, possibly harnessing other ideas from that domain could bring further benefits.

**Task decomposition & planning** increases the effectiveness of LLMs by dividing a task into subtasks. Examples include ADaPT (Prasad et al., 2024), ANPL (Huang et al., 2023), and others (Zhu et al., 2025; Shen et al., 2023). KGoT already harnesses *recursive* task decomposition: the input task is divided into numerous steps, and many of these steps are further decomposed by the LLM Graph Executor if necessary. For example, when solving a task based on the already constructed KG, the LLM Graph Executor may decide to decompose this step similarly to ADaPT. Other decomposition schemes could also be tried, we leave this as future work.

**Retrieval-Augmented Generation (RAG)** is an important part of the LLM ecosystem, with numerous designs being proposed (Edge et al., 2025; Gao et al., 2024; Besta et al., 2025b; Zhao et al., 2024b; Hu & Lu, 2025; Huang & Huang, 2024; Yu et al., 2024a; Mialon et al., 2023; Li et al., 2022; Abdallah & Jatowt, 2024; Delile et al., 2024; Manathunga & Illangasekara, 2023; Zeng et al., 2024; Wewer et al., 2021; Xu et al., 2024; Sarthi et al., 2024; Asai et al., 2024; Yu et al., 2024b; Gutiérrez et al., 2024). RAG has been used primarily to ensure data privacy and to reduce hallucinations. We illustrate that it has lower performance than KGoT when applied to AI assistant tasks.

## 7 CONCLUSION

In this paper, we introduce Knowledge Graph of Thoughts (KGoT), an AI assistant architecture that enhances the reasoning capabilities of low-cost models while significantly reducing operational expenses. By dynamically constructing and evolving knowledge graphs (KGs) that encode the task and its resolution state, KGoT enables structured knowledge representation and retrieval, improving task success rates on benchmarks such as GAIA and SimpleQA. Our extensive evaluation demonstrates that KGoT outperforms existing LLM-based agent solutions, for example achieving a substantial increase in task-solving efficiency of 29% or more over the competitive Hugging Face Agents baseline, while ensuring over $36\times$ lower costs. Thanks to its modular design, KGoT can be extended to new domains that require complex multi-step reasoning integrated with extensive interactions with the external compute environment, for example automated scientific discovery or software design.

**Ethics statement.** This work uses datasets and models that are publicly available or licensed for use. We did not conduct research involving human subjects and did not collect or share any personal or sensitive information. All data was used according to its license terms. Our work focuses on improving the quality of question answering with tool usage and therefore any risks that apply to this setting in general, also apply to our work. However our work does not create new risks.

**Reproducibility statement.** We make our results easy to reproduce. We release the code of our framework as well as self-implemented baselines, configuration files, and scripts to execute all evaluations, along with detailed instructions, hyperparameters, prompts, and random seeds. We document software and hardware versions and model/dataset identifiers. Where licensing prevents sharing specific datasets, we provide preprocessing scripts and exact instructions to reconstruct them. All figures and tables in the paper can be regenerated.

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

# A ADDITIONAL EXAMPLES OF KNOWLEDGE GRAPH REPRESENTATION OF TASKS

We include selected snapshots of KG representation of tasks, covering a wide range of graph structures from simple chains to trees and cyclic graphs. Each snapshot captures the current KG state in a JSON file, exported using a predefined query that retrieves all labeled nodes and edges. Regardless of the underlying graph backend, the use of a consistent export format allows all snapshots to be visualized through Neo4j's built-in web interface. In the following, we showcase illustrations of such snapshots and task statements from the GAIA validation set. Please note that the GAIA benchmark discourages making its tasks accessible to crawling. **To honor their wishes, we replaced the names of entities with placeholders in the following examples, while keeping the overall structure intact.**

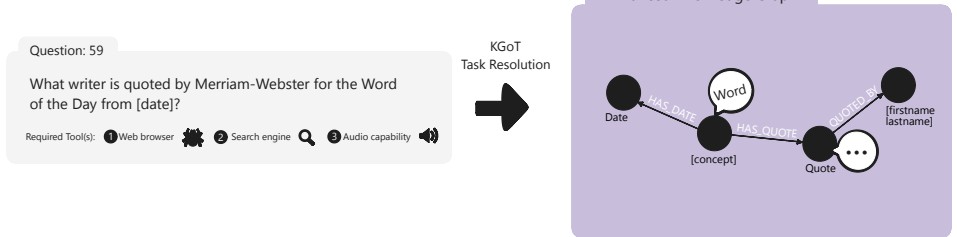

Figure 6: **Example of a chain structure.** This task requires 7 intermediate steps and the usage of 3 tools. The expected solution is '[firstname lastname]'. KGoT invokes the Surfer agent to search for relevant pages, locate the relevant quote, and find the person who said it. All intermediate information is successfully retrieved and used for enhancing the dynamically constructed KG. The quote contains two properties, significance and text. 'significance' stores the meaning of the quote, whereas 'text' stores the actual quote.

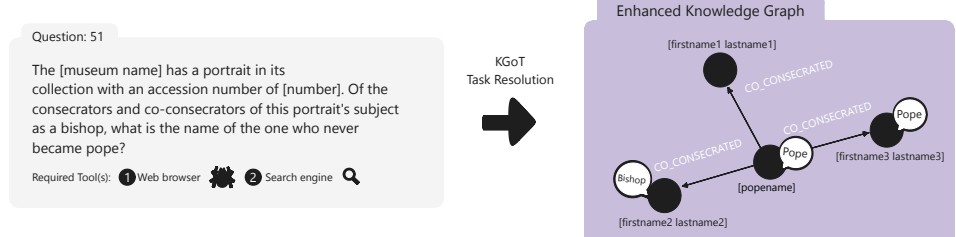

Figure 7: **Example of a tree structure.** This task requires 6 intermediate steps and the usage of 2 tools. The expected solution is '[firstname1 lastname1]'. The Surfer agent is also invoked for this task. In this KG representation of the task, [popename] is identified as the consecrator, where [firstname1 lastname1], [firstname2 lastname2] and [firstname3 lastname3] are all co-consecrators. Subsequently, the correct answer is obtained from the KGoT from the KG by correctly identifying [firstname1 lastname1] as the one without any labels.

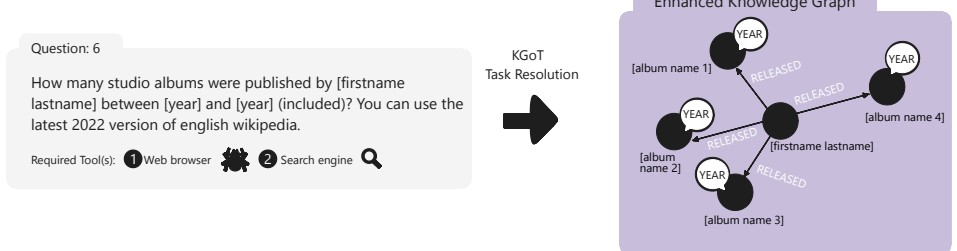

Figure 8: **Example of a tree structure.** This task requires 4 intermediate steps and the usage of 2 tools. The expected solution is '4'. This is a trap question where only the studio albums should be taken into account. In addition to years, the type of the albums is also stored as a property in the KG. Please note that the original GAIA task has a different solution, which we do not want to reveal.

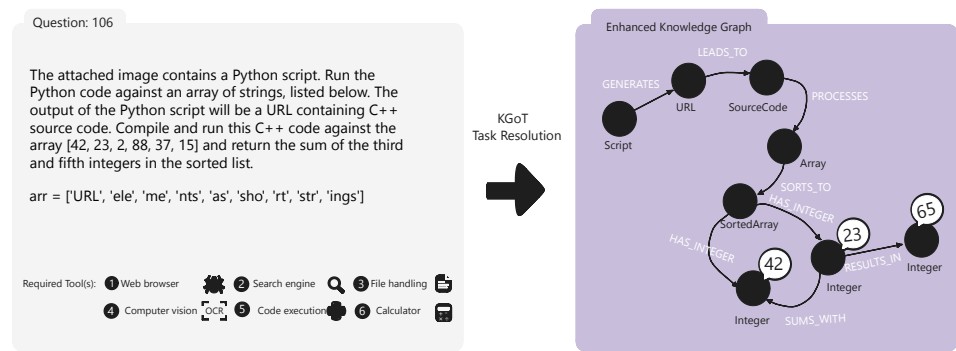

Figure 9: **Example of a cyclic graph structure.** This task requires 7 intermediate steps and the usage of 6 tools. The expected solution is '65'. Here, array has the property 'values' with [42, 23, 2, 88, 37, 15], SortedArray contains the correctly sorted values [2, 15, 23, 37, 42, 88]. The final solution '65' is correctly retrieved and parsed as KGoT response. Please note that we used different array values than in the original GAIA task.

## A.1 GRAPH STORAGE REPRESENTATION OF KNOWLEDGE GRAPH EXAMPLES

We now illustrate two examples of knowledge graphs and how they are represented in Neo4j and NetworkX respectively as well as the queries used to extract the final solution. Please note again, that we either replaced the values with placeholders (first question) or with different values (second question) in order to not leak the GAIA benchmark questions.

We start with GAIA question 59, which is illustrated in Figure 6. The knowledge graph stored in **Neo4j** after the first iteration is shown in the code snippet below.

---

**Neo4j KG representation while processing question 59.**

```
Nodes:
  Label: Writer
    {neo4j_id:0, properties:{'name': '[firstname lastname]'}}
  Label: WordOfTheDay
    {neo4j_id:1, properties:{'pronunciation': '[con-cept]', 'definition':
      'textual definition', 'counter': 1, 'origin': 'some war between year-year',
      'word': '[concept]', 'date': '[date1]'}}
  Label: Quote
    {neo4j_id:2, properties:{'text': '[quote]', 'source': '[newspaper name]',
      'date': '[date2]'}}
Relationships:
  Label: QUOTED_FOR
  {source: {neo4j_id: 0, label: Writer}, target: {neo4j_id: 1, label: WordOfTheDay},
      properties: {}}
  Label: QUOTED_IN
    {source: {neo4j_id: 0, label: Writer}, target: {neo4j_id: 2, label: Quote},
      properties: {}}
```

---

The Cypher query used to extract the solution was the following:

---

**Cypher query to extract the solution for question 59.**

```
MATCH (w:Writer)-[:QUOTED_FOR]->(wod:WordOfTheDay {date: '[date1]'})
RETURN w.name AS writer_name
```

---

To illustrate the use of **NetworkX**, we use a knowledge graph for question 6 (shown in Figure 8) from the GAIA benchmark after the second iteration.

**NetworkX KG representation while processing question 6.**

```
Existing Nodes:
 Label: Function
   [{id:A1, properties:{'name': 'image_inspector'}},
     {id:call_X2CcPnp5acMUPAp1Qx3OTvKx, properties:{'name': 'image_inspector',
     'args': {'question': 'What Python script is depicted in the attached image?',
     'full_path_to_image': '[filepath].png'}}}]
 Label: Script
  [{id:A2, properties:{'description': 'Python script to construct a URL by combining
     a base URL with specific indices from an array'}}]
 Label: Array
   [{id:A3, properties:{'content': "['URL', 'ele', 'me', 'nts', 'as', 'sho', 'rt',
     'str', 'ings']"}}]
 Label: URL
   [{id:A4, properties:{'base': '[base URL]', 'indices': [some indices]}}]

Existing Relationships:
 Label: uses
   [{source: {id: A1}, target: {id: A2}, properties: {}}]
 Label: contains
   [{source: {id: A2}, target: {id: A3}, properties: {}}]
 Label: constructs
   [{source: {id: A2}, target: {id: A4}, properties: {}}]
 Label: None
   [{source: {id: call_X2CcPnp5acMUPAp1Qx3OTvKx}, target: {id: A2}, properties: {}}]
```

The following Python code was used to extract the final solution:

**Python code to extract the solution for question 6.**

```python
# Retrieve the base URL and indices to construct the final URL
base_url = self.G.nodes['A4']['base']
indices = self.G.nodes['A4']['indices']

# Retrieve the array content
arr = eval(self.G.nodes['A3']['content'])

# Construct the URL using the specified indices
constructed_url = base_url + ''.join(arr[i] for i in indices)

# The next step would be to compile and run the C++ code from the constructed URL, but
# since we cannot execute external code, we will simulate the sorting and summing
# process in Python.

# Simulating the C++ code execution with the given array
sorted_arr = sorted([2, 15, 23, 37, 42, 88])
# Sum of the third and fifth integers in the sorted list
result = sorted_arr[2] + sorted_arr[4]
```

After the code execution, the correct solution of 65 is obtained.

# B  ADDITIONAL DETAILS ON SYSTEM DESIGN & IMPLEMENTATION

## B.1  GENERAL IMPLEMENTATION DETAILS

KGoT employs Docker (Docker Inc., 2025) and Sarus (Benedicic et al., 2019) for containerization, enabling a consistent and isolated runtime environment for all components. We containerize critical modules such as the KGoT controller, the Neo4j knowledge graph, and integrated tools (e.g., the Python Executor tool for safely running LLM-generated code with timeouts). Here, **Docker** provides a widely adopted containerization platform for local and cloud deployments that guarantees consistency between development and production environments. **Sarus**, a specialized container platform designed for high-performance computing (HPC) environments, extends KGoT's portability to HPC settings where Docker is typically unavailable due to security constraints. This integration allows KGoT to operate efficiently in HPC environments, leveraging their computational power.

KGoT also harnesses LangChain (LangChain Inc., 2025a), an open-source framework specifically designed for creating and orchestrating LLM-driven applications. LangChain offers a comprehensive suite of tools and APIs that simplify the complexities of managing LLMs, including prompt engineering, tool integration, and the coordination of LLM outputs.

## B.2  CONTROLLER

The Controller is the central orchestrator of the KGoT system, responsible for managing the interaction between the knowledge graph and the integrated tools. When a user submits a query, the Controller initiates the reasoning process by interpreting the task and coordinating the steps required for its resolution.

To offer fine-grained control over the KGoT control logic, the following parameters can be configured:

- `num_next_steps_decision`: Number of times to prompt an LLM on how to proceed (Solve/Enhance). Defaults to 5.

- `max_retrieve_query_retry`: Maximum retries for a Solve query when the initial attempt fails. Defaults to 3.

- `max_cypher_fixing_retry`: Maximum retries for fixing a Cypher query that encounter errors. Defaults to 3.

- `max_final_solution_parsing`: Maximum retries of parsing the final solution from the output of the Solve query. Defaults to 3.

- `max_tool_retries`: Maximum number of retries when a tool invocation fails. Defaults to 6.

Controller classes derived from the `ControllerInterface` abstract class embed such parameters with default values defined for their class. Users can experiment with custom parameters as well. We discuss how the choice of these parameters impacts the system robustness in Appendix B.3.

### B.2.1  ARCHITECTURE

The KGoT Controller employs a dual-LLM architecture with a clear separation of roles between constructing the knowledge graph (managed by the LLM Graph Executor) and interacting with tools (managed by the LLM Tool Executor). The following discussion provides additional specifics to the workflow description in Section 4.

The **LLM Graph Executor** is responsible for decision making and orchestrating the knowledge graph-based task resolution workflow, leading to different pathways (Solve or Enhance).

- `define_next_step`: Determine the next step. This function is invoked up to `num_next_steps_decision` times to collect replies from an LLM, which are subsequently used with a majority vote to decide whether to retrieve information from the knowledge graph for solving the task (Solve) or insert new information (Enhance).

- `_insert_logic`: Run Enhance. Once we have successfully executed tool calls and gathered new information, the system generates the Enhance query or queries to modify the knowledge graph accordingly. Each Enhance query is executed and its output is validated.

- `_retrieve_logic`: Run Solve. If the majority vote directs the system to the Solve pathway, a predefined solution technique (direct or query-based retrieve) is used for the solution generation.

- `_get_math_response`: Apply additional mathematical processing (optional).

- `parse_solution_with_llm`: Parse the final solution into a suitable format and prepare it as the KGoT response.

The **LLM Tool Executor** decides which tools to use as well as handling the interaction with these tools.

- `define_tool_calls`: Define tool calls. The system orchestrates the appropriate tool calls based on the knowledge graph state.

- `_invoke_tools_after_llm_response`, `_invoke_tool_with_retry`: Run tool calls with or without retry.

### B.3 Enhancing System Robustness

Given the non-deterministic nature of LLMs and their potential for generating hallucinations (Kaddour et al., 2023), the robustness of KGoT has been a fundamental focus throughout its design and implementation. Ensuring that the system consistently delivers accurate and reliable results across various scenarios is paramount. One of the key strategies employed to enhance robustness is the use of **majority voting**, also known as self-consistency (Wang et al., 2023b). In KGoT, majority voting is implemented by querying the LLM multiple times (by default 5 times) when deciding the next step, whether to insert more data into the knowledge graph or retrieve existing data. This approach reduces the impact of single-instance errors or inconsistencies, ensuring that the decisions made reflect the LLM's most consistent reasoning paths.

The choice of defaulting to five iterations for majority voting is a strategic balance between reliability and cost management, and was based on the work by Wang et al. (2023b), which showed diminishing returns beyond this point.

In addition, KGoT uses a separate default iteration count of seven for executing its full range of functions during problem-solving. These seven iterations correspond to the typical number of tool calls required to thoroughly explore the problem space, including multiple interactions with tools like the Surfer agent and the external LLM. Unlike the five iterations used for majority voting used to ensure robustness, this strategy ensures the system leverages its resources effectively across multiple tool invocations before concluding with a "No Solution" response if the problem remains unresolved.

**Layered Error-Checking:** KGoT integrates multiple error-checking mechanisms to safeguard against potential issues. The system continuously monitors for syntax errors and failures in API calls. These mechanisms are complemented by custom parsers and retry protocols. The parsers, customized from LangChain (LangChain Inc., 2025d), are designed to extract the required information from the LLM's responses, eliminating the need for manual parsing. Additionally, different decoders are used to handle cases where the LLM returns responses in various encodings . In cases where errors persist despite initial correction attempts, the system employs retry mechanisms. These involve the LLM rephrasing the Cypher queries and try them again. The Controller's design includes a limit on the number of retries for generating Cypher queries and invoking tools, balancing the need for error resolution with the practical constraints of time and computational resources. More information can be found in the subsequent section.

### B.4 Error Management Techniques

#### B.4.1 Handling LLM-Generated Syntax Errors

Syntax errors generated by LLMs can disrupt the workflow of KGoT, potentially leading to incorrect or incomplete solutions, or even causing the system to fail entirely. To manage these errors, KGoT includes LangChain's JSON parsers (LangChain Inc., 2025d) that detect syntax issues.

When a syntax error is detected, the system first attempts to correct it by adjusting the problematic syntax using different encoders, such as `"unicode_escape"` (Python Software Foundation, 2025a). If the issue persists, KGoT employs a retry mechanism that uses the LLM to rephrase the query/-command and attempts to regenerate its output. This retry mechanism is designed to handle up to

three attempts, after which the system logs the error for further analysis, bypasses the problematic query, and continues with other iterations in the hope that another tool or LLM call will still be able to resolve the problem.

A significant issue encountered with LLM-generated responses is managing the escape characters, especially when returning a Cypher query inside the standard JSON structure expected by the LangChain parser. The combination of retries using different encoders and parsers has mitigated the problem, though not entirely resolved it. Manual parsing and the use of regular expressions have also been attempted but with limited success.

### B.4.2 Managing API and System Errors

API-related errors, such as the OpenAI code '500' errors, are a common challenge in the operation of KGoT, especially when the external servers are overwhelmed. To manage these errors, the primary strategy employed is exponential backoff, which is a technique where the system waits for progressively longer intervals before retrying a failed API call, reducing the likelihood of repeated failures due to temporary server issues or rate limits (Tenacity Developers, 20245). In KGoT, this approach is implemented using the `tenacity` library, with a retry policy that waits for random intervals ranging from 1 to 60 seconds and allows for up to six retry attempts (`wait=wait_random_exponential(min=1, max=60)`, `stop=stop_after_attempt(6)`).

Additionally, KGoT includes comprehensive logging systems as part of its error management framework. These systems track the errors encountered during system operation, providing valuable data that can be easily parsed and analyzed (e.g. snapshots of the knowledge graphs or responses from third-party APIs). This data can then be used to refine the system's error-handling protocols and improve overall reliability.

It is also important to note that the system's error management strategies are built on top of existing errors systems provided by external tools, such as the LangChain interface for OpenAI, which already implements a default exponential backoff strategy with up to six retries (LangChain Inc., 2025b). These built-in mechanisms complement KGoT's own error-handling strategies, creating a multi-layered defense against potential failures and ensuring high levels of system reliability.

### B.5 Detailed Tool Description

Tools are a fundamental component of the KGoT framework, enabling seamless interaction with external resources such as the web and various file formats. KGoT currently supports the following tools:

- **Python Code Tool**: Executes code snippets provided by the LLM in a secure Python environment hosted within a Docker (or Sarus) container. This ensures that any potential security risks from executing untrusted code are mitigated. Besides running code, this tool is also utilized for mathematical computations.

- **Large Language Model (LLM) Tool**: Allows the LLM Tool Executor to request data generation from another instance of the same LLM. It is primarily employed for simple, objective tasks where no other tool is applicable.

- **Surfer Agent**: This web browser agent leverages SerpAPI to perform efficient Google searches and extract relevant webpage data. Built on Hugging Face Agents (Roucher & Petrov, 2025), this tool combines the capabilities with our WebCrawler and Wikipedia tools while adding support for JavaScript-rendered pages. It uses viewpoint segmentation to prevent the *"lost in the middle effect"* and incorporates additional navigation functionalities, such as search and page traversal.

- **ExtractZip Tool**: Extracts data from compressed files (e.g., ZIP archives). It was enhanced through integration with the TextInspector Tool, enabling seamless analysis of extracted files without requiring additional iterations to process the data.

- **TextInspector Tool**: A versatile tool for extracting data from multiple file types, including PDFs, spreadsheets, MP3s, and YouTube videos. It organizes extracted content in Markdown format, enhancing readability and integration into the Knowledge Graph. The tool was augmented with the best components from our original MultiModal Tool and the Hugging Face Agents TextInspector Tool. It can directly process questions about extracted content without returning the raw data to the LLM.

- **Image Tool**: Extracts information from images, such as text or objects, and returns it in a structured format. This tool is crucial for tasks requiring image processing and analysis. We selected the best prompts from our original tool set as well as Hugging Face Agents to optimize data extraction and analysis.

Tool integration within the KGoT framework is crucial for extending the system's problem-solving capabilities beyond what is achievable by LLMs alone. The strategy is designed to be modular, scalable, and efficient, enabling the system to leverage a diverse array of external tools for tasks such as data retrieval, complex computations, document processing, and more.

### B.5.1 Modular Tool Architecture

All tools integrated into the KGoT system are built upon the `BaseTool` abstraction provided by the LangChain framework (LangChain Inc., 2025c). This standardized approach ensures consistency and interoperability among different tools, facilitating seamless integration and management of new tools. Each tool implementation adheres to the following structure:

- **tool_name**: A unique identifier for the tool, used by the system to reference and invoke the appropriate functionality.

- **description**: A detailed explanation of the tool's purpose, capabilities, and appropriate usage scenarios. This description assists the LLM Tool Executor in selecting the right tool for specific tasks. Including few-shot examples is recommended, though the description must adhere to the 1024-character limit imposed by `BaseTool`.

- **args_schema**: A schema defining the expected input arguments for the tool, including their types and descriptions. This schema ensures that the LLM Tool Executor provides correctly formatted and valid inputs when invoking the tool.

This structured definition enables the LLM Tool Executor to dynamically understand and interact with a wide array of tools, promoting flexibility and extensibility within the KGoT system.

### B.5.2 Tool Management and Initialization

The **ToolManager** component is responsible for initializing and maintaining the suite of tools available to the KGoT system. It handles tasks such as loading tool configurations, setting up necessary environment variables (e.g., API keys), and conducting initial tests to verify tool readiness, such as checking whether the `RunPythonCodeTool`'s Docker container is running. The ToolManager ensures that all tools are properly configured and available for use during the system's operation.

---

**Simplified example of ToolManager initialization.**

```
class ToolManager:
    def __init__(self):
        self.set_env_keys()
        self.tools = [
            LLM_tool(...),
            image_question_tool(...),
            textInspectorTool(...),
            search_tool(...),
            run_python_tool(...),
            extract_zip_tool(...),
            # Additional tools can be added here
        ]
        self.test_tools()

    def get_tools(self):
        return self.tools
```

---

This modular setup allows for the easy addition or removal of tools, enabling the system to adapt to evolving requirements and incorporate new functionalities as needed.

### B.5.3 INFORMATION PARSING AND VALIDATION

After a tool executes and returns its output, the retrieved information undergoes a parsing and validation process by the LLM Graph Executor before being integrated into the knowledge graph. This process ensures the integrity and relevance of new data:

- **Relevance Verification**: The content of the retrieved information is assessed for relevance to the original problem context. This step may involve cross-referencing with existing knowledge, checking for logical consistency, and filtering out extraneous or irrelevant details. The LLM Graph Executor handles this during Cypher query generation.
- **Integration into Knowledge Graph**: Validated and appropriately formatted information is then seamlessly integrated into the knowledge graph by executing each Cypher query (with required error managements as mentioned in section B.4.1), enriching the system's understanding and enabling more informed reasoning in future iterations.

### B.5.4 BENEFITS

This structured and systematic approach to tool integration and selection offers several key benefits:

- **Enhanced Capability**: By leveraging specialized tools, KGoT can handle a wide range of complex tasks that go beyond the inherent capabilities of LLMs, providing more comprehensive and accurate solutions.
- **Scalability**: The modular architecture allows for easy expansion of the tool set, enabling the system to adapt to new domains and problem types with minimal reconfiguration.
- **Flexibility**: The system's ability to adaptively select and coordinate multiple tools in response to dynamic problem contexts ensures robust and versatile problem-solving capabilities.

### B.6 HIGH-PERFORMANCE & SCALABILITY

As previously discussed, we also experimented with various high-performance computing techniques adopted to accelerate KGoT. This section outlines additional design details.

The acceleration strategies can be classified into two categories: those targeting the speedup of a single task, and those aimed at accelerating the execution of KGoT on a batch of tasks such as the GAIA benchmark.

Optimizations in the first category are:

- **Asynchronous Execution**: Profiling of the KGoT workflow reveals that a substantial portion of runtime is spent on LLM model calls and tool invocations. As this represents a typical I/O-intensive workload, Python multi-threading is sufficient to address the bottleneck. KGoT dynamically schedules independent I/O operations (based on the current graph state and execution logic) using asyncio to achieve full concurrency.
- **Graph Operation Parallelism**: KGoT maintains a graph storage backend for managing the knowledge graph. When new knowledge is obtained from the tools, KGoT generates a list of queries, which represent a sequence of graph operations to add or modify nodes, properties, and edges. However, executing these operations sequentially in the graph storage backend can be time-consuming. A key observation is that many of these operations exhibit potential independence. We leveraged this potential parallelism to accelerate these graph storage operations. Our solution involves having KGoT request an LLM to analyze dependencies within the operations and return multiple independent chains of graph storage operations. These chains are then executed concurrently using the asynchronous method proposed earlier, enabling parallel execution of queries on the graph storage. This approach effectively harnesses the inherent parallelism to significantly improve processing speed.

The applied optimizations result in an overall speedup of 2.30× compared to the sequential baseline for a single KGoT task.

The second category focuses on accelerating a batch of tasks, for which **MPI-based distributed processing** is employed. Additional optimizations have also been implemented to further enhance performance.

- **Work Stealing**: The work-stealing algorithm operates by allowing idle processors to "steal" tasks from the queues of busy processors, ensuring balanced workload distribution. Each processor maintains its task queue, prioritizing local execution, while stealing occurs only when its queue is empty. This approach reduces idle time and enhances parallel efficiency. Our implementation of the work-stealing algorithm for KGoT adopts a novel approach tailored for distributed atomic task execution in an MPI environment. Each question is treated as an atomic task, initially distributed evenly across all ranks to ensure balanced workload allocation. When a rank completes all its assigned tasks, it enters a work-stealing phase, prioritizing the rank with the largest queue of remaining tasks. Operating in a peer-to-peer mode without a designated master rank, each rank maintains a work-stealing monitor to handle task redistribution. This monitor tracks incoming requests and facilitates the transfer of the last available task to the requesting rank whenever feasible. The system ensures continuous work-stealing, dynamically redistributing tasks to idle ranks, thus minimizing idle time and maximizing computational efficiency across all ranks. This decentralized and adaptive strategy significantly enhances the parallel processing capabilities of KGoT .

- **Container Pool**: The container pool implementation for KGoT ensures modular and independent execution of each tasks on separate ranks by running essential modules, such as Neo4j and the Python tool, within isolated containers, with one container assigned per rank. We use a Kubernetes-like container orchestration tool specifically designed for KGoT running with MPI. The container pool supports Docker and Sarus to be compatible with local and cluster environments. Our design guarantees that each task operates independently without interfering with each other, while trying to minimize latency between the KGoT controller and the containers.

Ultimately, our experiments achieved a $12.74\times$ speedup over the sequential baseline on the GAIA benchmark when executed with 8 ranks in MPI, as illustrated in Figure 10. This demonstrates the significant performance improvement of the KGoT system achieved on a consumer-grade platform.

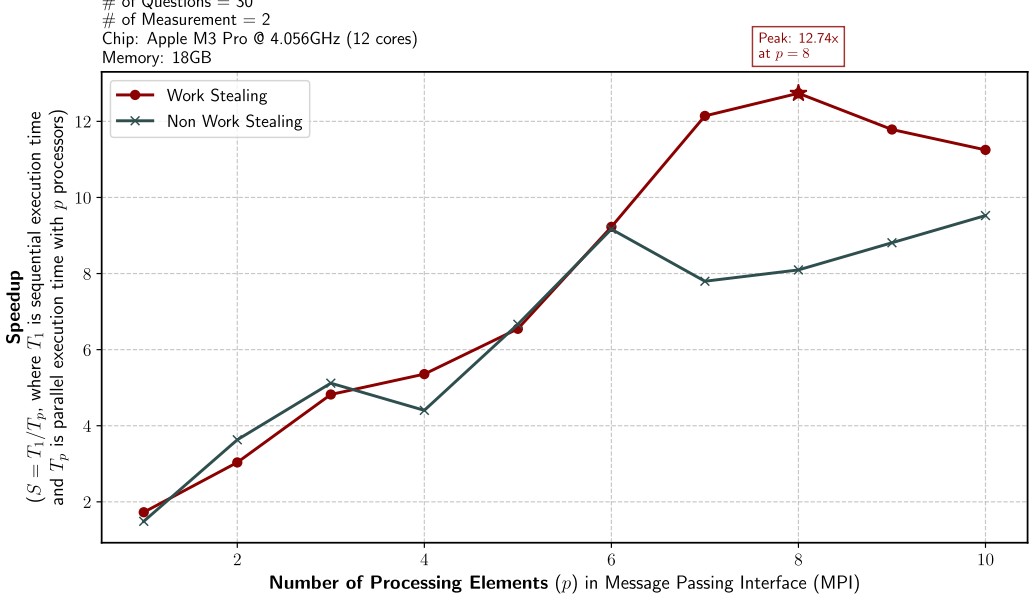

Figure 10: Measured parallel speedup of KGoT task execution across varying numbers of MPI processes, under two scheduling strategies: with and without work stealing. Each task corresponds to a GAIA benchmark question, and each data point represents the average of 2 measurements on an Apple M3 Pro (12 cores @ 4.056GHz) and 18GB Memory. The dashed grey line indicates the expected theoretical speedup curve ($S = 2.2985 \times p$) based on the asynchronous optimizations applied to individual tasks. As previously discussed, acceleration strategies are categorized into (1) single-task optimizations—including asynchronous I/O scheduling and graph operation parallelism—and (2) batch-level parallelism using MPI-based distributed processing. The work-stealing variant consistently outperforms the non-stealing baseline by minimizing idle time and dynamically redistributing atomic question tasks across ranks. These combined strategies result in a $12.74\times$ speedup over the sequential baseline when using 8 processes.

## B.7 EXAMPLES OF NOISE MITIGATION

We illustrate two examples of experiments with noise mitigation in KGoT. As before, we have replaced the specific values with placeholders to prevent the leakage of the GAIA benchmark tasks.

### B.7.1 IRRELEVANCE REMOVAL

The first example is based on question 146 in the validation set of the GAIA benchmark:

*On [date], an article by [author] was published in [publication]. This article mentions a team that produced a paper about their observations, linked at the bottom of the article. Find this paper. Under what NASA award number was the work performed by [researcher] supported by?*

The example KG has been populated with data directly related to the answer as well as information that is relevant to the question but not necessary for answering it. Removing this extraneous data makes it easier for KGoT to reason about the KG content and extract data relevant to the answer. The data to be removed is marked in red.

---

**Question 146: Initial state of the knowledge graph.**

```
Nodes:
Label: Funding
{neo4j_id:0, properties:{'award_number': '[award_number]'}}
Label: Researcher
{neo4j_id:13, properties:{'name': '[researcher]'}}
Label: Article
{neo4j_id:11, properties:{'author': '[author]', 'title': '[title]', 'source':
'[publication]', 'publication_date': '[date]'}}
Label: Paper
{neo4j_id:12, properties:{'title': '[paper]'}}

Relationships:
Label: SUPPORTED_BY
{source: {neo4j_id: 13, label: Researcher},
target: {neo4j_id: 0, label: Funding}, properties: {}}
Label: LINKED_TO
{source: {neo4j_id: 11, label: Article}, target: {neo4j_id: 12, label: Paper},
properties: {}}
Label: INVOLVES
{source: {neo4j_id: 12, label: Paper}, target: {neo4j_id: 13, label: Researcher},
properties: {}}
```

---

**Question 146: Denoised knowledge graph.**

```
Nodes:
Label: Funding
{neo4j_id:0, properties:{'award_number': '[award_number'}}
Label: Researcher
{neo4j_id:13, properties:{'name': '[researcher]'}}

Relationships:
Label: SUPPORTED_BY
{source: {neo4j_id: 13, label: Researcher},
target: {neo4j_id: 0, label: Funding}, properties: {}}
```

---

### B.7.2 DUPLICATE REMOVAL

The second example is based on question 25 in the validation set of the GAIA benchmark:

*I need to fact-check a citation. This is the citation from the bibliography: [citation1]*
*And this is the in-line citation:*
*Our relationship with the authors of the works we read can often be "[quote]" ([citation2]).*

*Does the quoted text match what is actually in the article? If Yes, answer Yes, otherwise, give me the word in my citation that does not match with the correct one (without any article).*

In the example, the knowledge graph has been populated by two nearly identical nodes. The nodes and relationships marked for removal are shown in red.

**Question 25: Initial state of the knowledge graph.**

```
Nodes:
Label: Quote
{neo4j_id:22, properties:{'text': '[quote]'}}
{neo4j_id:0, properties:{'text': '[near_identical_quote]'}}

Label: Article
{neo4j_id:3, properties:{'journal': '[journal]',
'page_start': [page_start],
'author': '[author]',
'page_end': [page_end],
'title': '[title]',
'issue': [issue],
'volume': [volume],
'year': [year],
'doi': '[year]'}}

Relationships:
Label: CONTAINS
{source: {neo4j_id: 3, label: Article},
target: {neo4j_id: 22, label: Quote}, properties: {}}
{source: {neo4j_id: 3, label: Article},
target: {neo4j_id: 0, label: Quote}, properties: {}}
```

**Question 25: Denoised knowledge graph.**

```
Nodes:
Label: Quote
{neo4j_id:22, properties:{'text': '[quote]'}}

Label: Article
{neo4j_id:3, properties:{'journal': '[journal]',
'page_start': [page_start],
'author': '[author]',
'page_end': [page_end],
'title': '[title]',
'issue': [issue],
'volume': [volume],
'year': [year],
'doi': '[year]'}}

Relationships:
Label: CONTAINS
{source: {neo4j_id: 3, label: Article},
target: {neo4j_id: 22, label: Quote}, properties: {}}
```

## C  ADDITIONAL DETAILS ON PROMPT ENGINEERING

The primary objectives in our prompt design include improving decision-making processes, effectively managing complex scenarios, and allowing the LLM to adapt to diverse problem domains while maintaining high accuracy and efficiency. To achieve this, we leverage prompt engineering techniques, particularly the use of generic few-shot examples embedded in prompt templates. These examples guide the LLM in following instructions step by step (chain-of-thought) and reducing errors in generating graph queries with complex syntax.

### C.1  PROMPT FOR MAJORITY VOTING

At the beginning of each iteration, the LLM Graph Executor uses the following prompt to decide whether the task can be solved with the current KG or if more information is needed. For system robustness, it is run multiple times with varying reasoning paths, and a majority vote (**self-consistency**) is applied to the responses. The prompt also explicitly instructs the model to decide on either the Solve or the Enhance pathway. By requiring the model to output an indicator (query_type = "RETRIEVE" or "INSERT"), we can programmatically branch the workflow allowing for **control of reasoning pathways**.

---

**Graph Executor: Determine the next step**

```
<task>
You are a problem solver using a Neo4j database as a knowledge graph to solve a
given problem. Note that the database may be incomplete.
</task>

<instructions>
Understand the initial problem, the initial problem nuances, *ALL the existing
data* in the database and the tools already called. Can you solve the initial
problem using the existing data in the database?

    • If you can solve the initial problem with the existing data currently in
      the database return the final answer and set the query_type to RETRIEVE.
      Retrieve only if the data is sufficient to solve the problem in a zero-shot
      manner.

    • If the existing data is insufficient to solve the problem, return why you
      could not solve the initial problem and what is missing for you to solve
      it, and set query_type to INSERT.

    • Remember that if you don't have ALL the information requested, but only
      partial (e.g. there are still some calculations needed), you should continue
      to INSERT more data.
</instructions>

<examples>
<examples_retrieve>
<!-- In-context few-shot examples -->
</examples_retrieve>
<examples_insert>
<!-- In-context few-shot examples -->
</examples_insert>
</examples>

<initial_problem>
{initial_query}
</initial_problem>

<existing_data>
{existing_entities_and_relationships}
</existing_data>

<tool_calls_made>
{tool_calls_made}
</tool_calls_made>
```

---

## C.2 PROMPTS FOR ENHANCE PATHWAY

If the majority voting deems the current knowledge base as "insufficient", we enter the **Enhance Pathway**. To identify the **knowledge gap**, a list of reasons why the task is not solvable and what information is missing is synthesized by the LLM Graph Executor to a single, consistent description.

---

**Graph Executor: Identify missing information**

```
<task>
You are a logic expert, your task is to determine why a given problem cannot be
solved using the existing data in a Neo4j database.
</task>

<instructions>
You are provided with a list of reasons. Your job is to combine these reasons into
a single, coherent paragraph, ensuring that there are no duplicates.
    • Carefully review and understand each reason provided.
    • Synthesize the reasons into one unified text.
</instructions>

<list_of_reasons>
{list_of_reasons}
</list_of_reasons>
```

---

By providing both the current graph state and the identified missing information, the LLM Tool Executor defines **context-aware** tool calls to bridge the knowledge gap identified by the LLM Graph Executor.

---

**Tool Executor: Define tool calls**

```
<task>
You are an information retriever tasked with populating a Neo4j database with the
necessary information to solve the given initial problem.
</task>

<instructions>
<!-- In-context few-shot examples covering the following aspects:
1. **Understand Requirements**
2. **Gather Information**
3. **Detailed Usage**
4. **Utilize Existing Data**
5. **Avoid Redundant Calls**
6. **Ensure Uniqueness of Tool Calls**
7. **Default Tool**
8. **Do Not Hallucinate**-->
 </instructions>

<initial_problem>
{initial_query}
</initial_problem>

<existing_data>
{existing_entities_and_relationships}
</existing_data>

<missing_information>
{missing_information}
</missing_information>

<tool_calls_made>
{tool_calls_made}
</tool_calls_made>
```

---

Afterwards specialized tools such as a web browser or code executor are invoked to perform data retrieval from external resources. The newly acquired information is then used to enhance the KG. The LLM Graph Executor is asked to analyze the retrieved information in the context of the initial user query and the current state of the KG. The following prompt is carefully designed to guide the LLM to generate semantically correct and context-aware Cypher queries with concrete examples.

---

**Graph Executor: Create Cypher for data ingestion**

```
<task>
You are a problem solver tasked with updating an incomplete Neo4j database used
as a knowledge graph. You have just acquired new information that needs to be
integrated into the database.
</task>

<instructions>
<!-- In-context few-shot examples covering following aspects:
0. **Understand the Context**
1. **Use Provided New Information Only**
2. **No Calculations**
3. **Avoid Duplicates**
4. **Combine Operations with WITH Clauses**
5. **Group Related Queries**
6. **Omit RETURN Statements**
7. **Omit ID Usage**
8. **Merge Existing Nodes**
9. **Correct Syntax and Semantics**
10. **Use Correct Relationships**
11. **Escape Characters** -->
</instructions>

<initial_problem>
{initial_query}
</initial_problem>

<existing_data>
{existing_entities_and_relationships}
</existing_data>

<missing_information>
{missing_information}
</missing_information>

<new_information>
{new_information}
</new_information>
```

---

## C.3 Prompts for Solve Pathway

If majority voting confirms that the KG is sufficiently populated or the maximum iteration count has been reached, the system proceeds to the **Solve Pathway**. The iteratively refined KG serves as a reliable information source for LLMs to solve the initial query. To provide a robust response, we introduced two approaches, a **query-based** approach and **Direct Retrieval**, for knowledge extraction.

### C.3.1 Graph Query Language for Knowledge Extraction

The query-based approach formulates a read query using an LLM, given the entire graph state and other relevant information such as the initial problem. The LLM-generated query is then executed on the graph database to return the final solution. Please note KGoT iteratively executes the solve operations collected from the majority voting.

---

**In-context few-shot examples for query-based knowledge extraction**

```
<examples_retrieve>
<example_retrieve_1>
Initial problem: Retrieve all books written by ''J.K. Rowling''.

Existing entities:
Author: [{{name: ''J.K. Rowling'', author_id: ''A1''}, {{name: ''George
R.R. Martin'', author_id: ''A2''}}], Book: [{{title: ''Harry Potter and the
Philosopher's Stone'', book_id: ''B1''}, {{title: ''Harry Potter and the Chamber
of Secrets'', book_id: ''B2''}, {{title: ''A Game of Thrones'', book_id: ''B3''}}]

Existing relationships:
(A1)-[:WROTE]->(B1),
(A1)-[:WROTE]->(B2),
(A2)-[:WROTE]->(B3)

Solution:
query: '
MATCH (a:Author {{name: ''J.K. Rowling''}})-[:WROTE]->(b:Book)
RETURN b.title AS book_title'
query_type: RETRIEVE
</example_retrieve_1>

<example_retrieve_2>
Initial problem: List all colleagues of ''Bob''.
Existing entities: Employee: [{{name: ''Alice'', employee_id: ''E1''}, {{name:
''Bob'', employee_id: ''E2''}, {{name: ''Charlie'', employee_id: ''E3''}}],
Department: [{{name: ''HR'', department_id: ''D1''}, {{name: ''Engineering'',
department_id: ''D2''}}]

Existing relationships:
(E1)-[:WORKS_IN]->(D1),
(E2)-[:WORKS_IN]->(D1),
(E3)-[:WORKS_IN]->(D2)

Solution:
query: '
MATCH (e:Employee {name: "Bob"})-[:WORKS_IN]->(d:Department)
<-[:WORKS_IN]-(colleague:Employee)
WHERE colleague.name <> "Bob"
RETURN colleague.name AS colleague_name
'
query_type: RETRIEVE
</example_retrieve_2>
</examples_retrieve>
```

If the attempt to fix a previously generated query fails or the query did not return any results, KGoT will try to regenerate the query from scratch by providing the initial problem statement, the existing data as well as additionally the incorrect query.

---

**Graph Executor: Regeneration of Cypher query for data retrieval**

```
<task>
You are a problem solver expert in using a Neo4j database as a knowledge graph.
Your task is to solve a given problem by generating a correct Cypher query. You
will be provided with the initial problem, existing data in the database, and a
previous incorrect Cypher query that returned an empty result. Your goal is to
create a new Cypher query that returns the correct results.
</task>

<instructions>
    1. Understand the initial problem, the problem nuances and the existing data
       in the database.

    2. Analyze the provided incorrect query to identify why it returned an empty
       result.

    3. Write a new Cypher query to retrieve the necessary data from the database
       to solve the initial problem. You can use ALL Cypher/Neo4j functionalities.

    4. Ensure the new query is accurate and follows correct Cypher syntax and
       semantics.

</instructions>

<examples>
<!-- In-context few-shot examples -->
</examples>

<initial_problem>
{initial_query}
</initial_problem>

<existing_data>
{existing_entities_and_relationships}
</existing_data>

<wrong_query>
{wrong_query}
</wrong_query>
```

### C.3.2 DIRECT RETRIEVAL FOR KNOWLEDGE EXTRACTION

Direct Retrieval refers to directly asking the LLM to formulate the final solution, given the entire graph state, without executing any LLM-generated read queries on the graph storage.

---

**In-context few-shot examples for DR-based knowledge extraction**

```
<examples_retrieve>
<example_retrieve_1>
Initial problem: Retrieve all books written by ``J.K. Rowling''.

Existing entities:
Author: [{{name: ``J.K. Rowling'', author_id: ``A1''}, {{name: ``George
R.R. Martin'', author_id: ``A2''}}], Book: [{{title: ``Harry Potter and the
Philosopher's Stone'', book_id: ``B1''}, {{title: ``Harry Potter and the Chamber
of Secrets'', book_id: ``B2''}, {{title: ``A Game of Thrones'', book_id: ``B3''}}]

Existing relationships:
(A1)-[:WROTE]->(B1),
(A1)-[:WROTE]->(B2),
(A2)-[:WROTE]->(B3)

Solution:
query: 'Harry Potter and the Philosopher's Stone, Harry Potter and the Chamber of
Secrets'
query_type: RETRIEVE
</example_retrieve_1>

<example_retrieve_2>
Initial problem: List all colleagues of ``Bob''.

Existing entities:
Employee: [{{name: ``Alice'', employee_id: ``E1''}, {{name: ``Bob'', employee_id:
``E2''}, {{name: ``Charlie'', employee_id: ``E3''}}], Department: [{{name: ``HR'',
department_id: ``D1''}, {{name: ``Engineering'', department_id: ``D2''}}]

Existing relationships:
(E1)-[:WORKS_IN]->(D1),
(E2)-[:WORKS_IN]->(D1),
(E3)-[:WORKS_IN]->(D2)

Solution:
query: 'Alice'
query_type: RETRIEVE
</example_retrieve_2>
</examples_retrieve>
```

### C.3.3 FORMATTING FINAL SOLUTION

After successful knowledge extraction from the KG, we obtain a partial answer to our initial query. Next, we examine if further post-processing, such as intermediate calculation or formatting, needs to be performed. In the following prompt, we first detect if any unresolved calculation is required.

---

**Solution formatting: Examine need for mathematical processing**

```
<task>
You are an expert in identifying the need for mathematical or probabilistic
calculations in problem-solving scenarios. Given an initial query and a partial
solution, your task is to determine whether the partial solution requires further
mathematical or probabilistic calculations to arrive at a complete solution. You
will return a boolean value: True if additional calculations are needed and False
if they are not.
</task>

<instructions>
    • Analyze the initial query and the provided partial solution.
    • Identify any elements in the query and partial solution that suggest
      the further need for numerical analysis, calculations, or probabilistic
      reasoning.
    • Consider if the partial solution includes all necessary numerical results
      or if there are unresolved numerical aspects.
    • Return true if the completion of the solution requires more calculations,
      otherwise return false.
    • Focus on the necessity for calculations rather than the nature of the math
      or probability involved.
</instructions>

<examples>
<!-- In-context few-shot examples -->
</examples>

<initial_problem>
{initial_query}
</initial_problem>

<partial_solution>
{partial_solution}
</partial_solution>
```

---

If any further mathematical processing is needed, the **Python Code Tool** is invoked to refine the current partial solution by executing an LLM-generated Python script. This ensures accuracy by leveraging the strength of LLMs in scripting. Moreover, it effectively avoids hallucinations by grounding outputs through verifiable and deterministic code computation.

---

**Solution formatting: Apply additional mathematical processing**

```
<task>
You are a math and python expert tasked with solving a mathematical problem.
</task>

<instructions>
To complete this task, follow these steps:
1. **Understand the Problem**:
      • Carefully read and understand the initial problem and the partial solution.
      • Elaborate on any mathematical calculations from the partial solution that
        are required to solve the initial problem.
2. **Perform Calculations**:
      • Use the run_python_code Tool to perform any necessary mathematical
        calculations.
      • Craft Python code that accurately calculates the required values based on
        the partial solution and the initial problem.
      • Remember to add print statements to display the reasoning behind the
        calculations.
      • **ALWAYS** add print statement for the final answer.
3. **Do Not Hallucinate**:
      • **Do not invent information** that is not provided in the initial problem
        or the partial solution.
      • **Do not perform calculations manually**; use the run_python_code Tool for
        all mathematical operations.
</instructions>

<initial_problem>
{initial_query}
</initial_problem>

<partial_solution>
{current_solution}
</partial_solution>
```

---

To produce a single, consistent answer and format the final solution to the initial user query, we guide the LLM with a dedicated prompt.

---

**Solution formatting: Parse the final solution**

```
<task>
You are a formatter and extractor. Your task is to combine partial solution from a
database and format them according to the initial problem statement.
</task>

<instructions>
    1. Understand the initial problem, the problem nuances, the desired output,
       and the desired output format.

    2. Review the provided partial solution.

    3. Integrate and elaborate on the various pieces of information from the
       partial solution to produce a complete solution to the initial problem. Do
       not invent any new information.

    4. Your final answer should be a number OR as few words as possible OR a comma
       separated list of numbers and/or strings.

    5. ADDITIONALLY, your final answer MUST adhere to any formatting instructions
       specified in the original question (e.g., alphabetization, sequencing,
       units, rounding, decimal places, etc.)

    6. If you are asked for a number, express it numerically (i.e., with digits
       rather than words), don't use commas, do not round the number unless
       directly specified, and DO NOT INCLUDE UNITS such as $ or USD or percent
       signs unless specified otherwise.

    7. If you are asked for a string, don't use articles or abbreviations (e.g.
       for cities), unless specified otherwise. Don't output any final sentence
       punctuation such as '.', '!', or '?'.

    8. If you are asked for a comma separated list, apply the above rules depending
       on whether the elements are numbers or strings.
</instructions>

<examples>
<!-- In-context few-shot examples -->
</examples>

<initial_problem>
{initial_query}
</initial_problem>

<given_partial_solution>
{partial_solution}
</given_partial_solution>
```

## C.4 PROMPT FOR LLM-GENERATED SYNTAX ERROR

In order to handle LLM-generated syntax errors, a retry mechanism is deployed to use the LLM to reformulate the graph query or code snippet, guided by specialized prompts tailored to the execution context. For Python code, the prompt guides the model to fix the code and update dependencies if needed, ensuring successful execution.

---

**Error handling: Fix invalid Python code**

```
<task>
You are an expert Python programmer. You will be provided with a block of Python
code, a list of required packages, and an error message that occurred during code
execution. Your task is to fix the code so that it runs successfully and provide
an updated list of required packages if necessary.
</task>

<instructions>
    1. Carefully analyze the provided Python code and the error message.

    2. Identify the root cause of the error.

    3. Modify the code to resolve the error.

    4. Update the list of required packages if any additional packages are needed.

    5. Ensure that the fixed code adheres to best practices where possible.
</instructions>

<rules>
    • You must return both the fixed Python code and the updated list of required
      packages.
    • Ensure the code and package list are in proper format.
</rules>

<examples>
<!-- In-context few-shot examples -->
</examples>


{code}


<required_modules>
{required_modules}
</required_modules>

<error>
{error}
</error>
```

---

For Cypher queries, the prompt helps the model diagnose syntax or escaping issues based on the error log and returns a corrected version.

---

**Error handling: Fix invalid Cypher query**

```
<task>
You are a Cypher expert, and you need to fix the syntax and semantic of a given
incorrect Cypher query.
</task>

<instructions>
Given the incorrect Cypher and the error log:

    1. Understand the source of the error (especially look out for wrongly
       escaped/not escaped characters).

    2. Correct the Cypher query

    3. Return the corrected Cypher query.

</instructions>

<wrong_cypher>
{cypher_to_fix}
</wrong_cypher>

<error_log>
{error_log}
</error_log>
```

---

Both prompts are reusable across pathways and enforce minimal, well-scoped corrections grounded in the provided error context.

# D   ADDITIONAL RESULTS

We plot the results from Figure 3 also as a Pareto front in Figure 11.

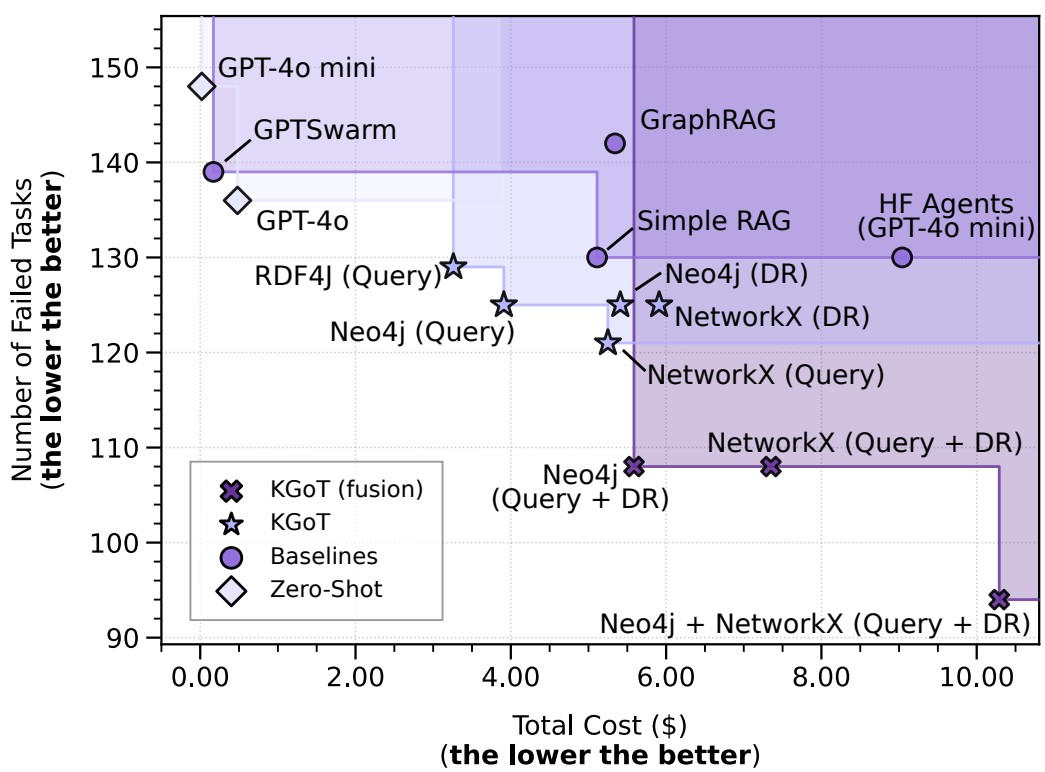

Figure 11: **Pareto front plot of cost and error counts.** We report results for answering 165 GAIA validation questions across different comparison targets, using the GPT-4o mini model with each baseline. For the Zero-Shot inference, we also include results for GPT-4o for comparison. Please note that we omit the results for Magentic-One and HF Agents (GPT-4o) as their high costs would heavily disturb the plot. DR means Direct Retrieval.

We also plot the relative improvements of KGoT over Hugging Face Agents and GPTSwarm respectively in Figure 12, which is based on the results shown in Figure 4.

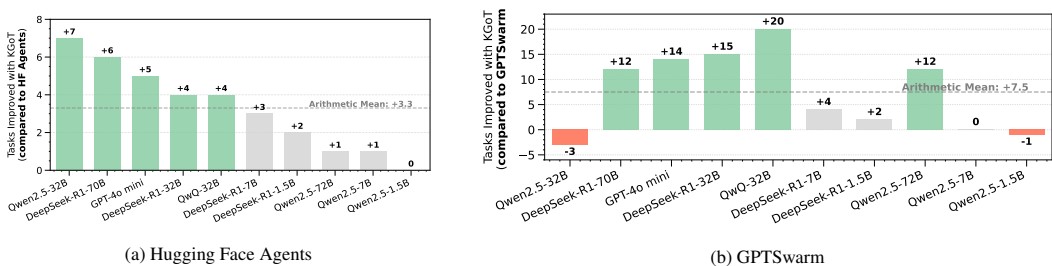

Figure 12: **Relative improvement of KGoT over Hugging Face Agents (left) and GPTSwarm (right) on the GAIA validation set using various LLM models.**

## D.1 SIMPLEQA RESULTS

Table 2: **Comparison of KGoT, HF Agents and GPTSwarm on a subset of SimpleQA as well as the results for KGoT on the full benchmark.** We highlight the best performing scheme in given category in bold. Model: GPT-4o mini.

| Framework | Correct (%) | Not attempted (%) | Incorrect (%) | Correct given attempted (%) | F-score | Total cost ($) | Cost per solved task ($) |
|---|---|---|---|---|---|---|---|
| GPTSwarm | 53.8106 | 6.2356 | 39.9538 | 57.3892 | 55.5 | **0.2159** | **0.00092660** |
| HF Agents | 66.0508 | 18.0139 | **15.9353** | **80.5634** | 72.6 | 16.7117 | 0.05843265 |
| KGoT | **73.2102** | **1.6166** | 25.1732 | 74.4131 | **73.8** | 5.6432 | 0.01780182 |
| KGoT (Full) | 70.3421 | 2.0342 | 27.8548 | 71.8027 | 71.1 | 59.1538 | 0.01943931 |

Table 3: **F1-score comparison of KGoT, OpenAI and Claude models on SimpleQA.** OpenAI and Claude results were taken from the official repository (OpenAI, 2025). Model for KGoT: GPT-4o mini.

| Reasoning Models | F1-score | Assistant Models | F1-score |
|---|---|---|---|
| o1 | 42.6 | gpt-4.1-2025-04-14 | 41.6 |
| o1-preview | 42.4 | gpt-4.1-mini-2025-04-14 | 16.8 |
| o3-high | 48.6 | gpt-4.1-nano-2025-04-14 | 7.6 |
| o3 | 49.4 | gpt-4o-2024-11-20 | 38.8 |
| o3-low | 49.4 | gpt-4o-2024-08-06 | 40.1 |
| o1-mini | 7.6 | gpt-4o-2024-05-13 | 39.0 |
| o3-mini-high | 13.8 | gpt-4o-mini-2024-07-18 | 9.5 |
| o3-mini | 13.4 | gpt-4.5-preview-2025-02-27 | 62.5 |
| o3-mini-low | 13.0 | gpt-4-turbo-2024-04-09 | 24.2 |
| o4-mini-high | 19.3 | Claude 3.5 Sonnet | 28.9 |
| o4-mini | 20.2 | Claude 3 Opus | 23.5 |
| o4-mini-low | 20.2 | | |
| **KGoT** | **71.1** | | |

## D.2 Impact from Various Design Decisions

Table 4: **Analysis of different design decisions and tool sets in KGoT.** "**ST**" stands for the type of the solve operation and pathway ("**GQ**": graph query, "**DR**": Direct Retrieval), "**PF**" for the prompt format ("**MD**": Markdown) and "**merged**" stands for a combination of the original KGoT tools and the Hugging Face Agents tools.

| Configuration | | | Metrics | | |
|---|---|---|---|---|---|
| Tools | ST | PF | Solved | Time (h) | Cost |
| HF | DR | XML | 37 | 11.87 | $7.84 |
| HF | GQ | MD | 33 | 9.70 | $4.28 |
| merged | GQ | XML | 31 | 10.62 | $5.43 |
| HF | GQ | XML | 30 | 13.02 | $4.90 |
| original KGoT | GQ | XML | 27 | 27.57 | $6.85 |

We explored **different tool sets**, with selected results presented in Table 4. Initially, we examined the limitations of our original tools and subsequently integrated the complete Hugging Face Agents tool set into the KGoT framework, which led to improvements in accuracy, runtime, and cost efficiency. A detailed analysis allowed us to merge the most effective components from both tool sets into an optimized hybrid tool set, further enhancing accuracy and runtime while only moderately increasing costs. Key improvements include a tighter integration between the ExtractZip tool and the Text Inspector tool, which now supports Markdown, as well as enhancements to the Surfer Agent, incorporating a Wikipedia tool and augmenting viewpoint segmentation with full-page summarization. This optimized tool set was used for all subsequent experiments.

We further evaluated **different prompt formats** in the initial iterations of KGoT. While our primary format was XML-based, we conducted additional tests using Markdown. Initial experiments with the Hugging Face Agents tool set (see Table 4) combined with Markdown and GPT-4o mini yielded improved accuracy, reduced runtime, and lower costs. However, these results were not consistently reproducible with GPT-4o. Moreover, Markdown-based prompts interfered with optimizations such as Direct Retrieval, ultimately leading us to retain the XML-based format.

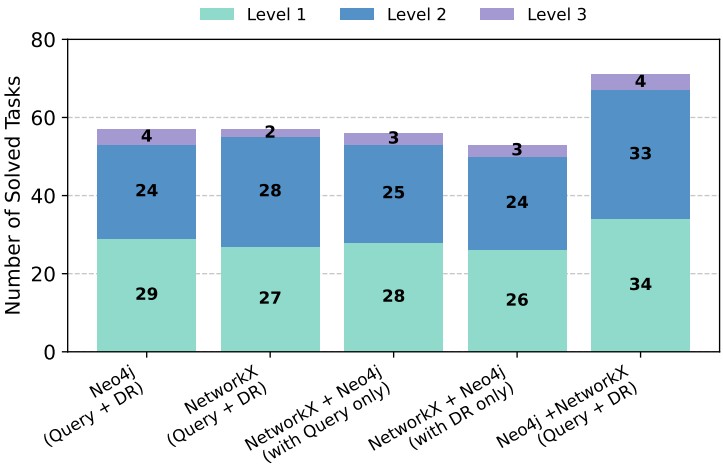

Figure 13: **Comparison of different fusion types in respect to the task solve operation as well as the graph backend type.** We report results for answering 165 GAIA validation questions across different comparison targets. DR stands for Direct Retrieval. Model: GPT-4o mini.

**Graph Backend vs. Task Solve Operation** We provide more detailed results in Figure 13, studying the performance of the following configurations: NetworkX + Neo4j (with query only) and NetworkX + Neo4j (with DR only) as well as Neo4j (query + DR) and NetworkX (query + DR). Overall, the fusion of backends (with DR only) offers smaller advantages than other types of fusion. This indicates that different graph querying languages have different strengths and their fusion comes with the largest combined advantage.

### D.3 Runtime

We provide a runtime overview of running KGoT on the validation set of the GAIA benchmark with GPT4o-mini, Neo4j and query-based retrieval in Figure 14. The right part follows the categorization in Appendix C. We provide a more detailed analysis of the runtime in Figure 17.

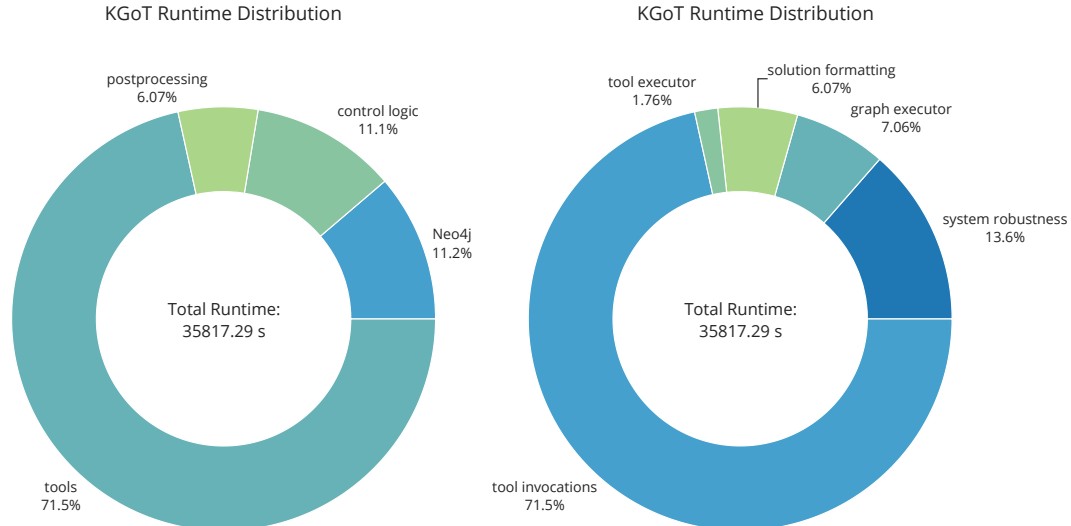

Figure 14: **Different runtime categorizations of the same data. Graph storage: Neo4j. Retrieval type: query. Model: GPT-4o mini.**

## D.4   COMPUTE RESOURCES

Because of the long runtime, we executed most experiments using the OpenAI API as an external resource on server compute nodes containing a AMD EPYC 7742 CPU with 128 cores running at 2.25GHz, with a total memory of 256GB. However when the LLM is called as an external resource, KGoT is able to run on commodity hardware with minimal effects on runtime.

Our experiments with locally run LLMs were executed with compute nodes containing 4x NVIDIA GH200, a respective GPU memory of 96GB, and a total memory of 896GB. In these cases, the minimum hardware requirements are dictated by the resources needed to run each LLM locally.

High-performance & scalability experiments were performed on an Apple M3 Pro with 12 cores at 4.056GHz and a total memory of 18GB.

## D.5   GAIA RESULT VISUALIZATIONS

We also implemented various automatic scripts that plot various aspects once a GAIA run is finished. In the following we provide example plots for Neo4j with query retrieval.

We provide a breakdown for each level of the GAIA benchmark into the categories that KGoT's answers for the tasks fall into in Figure 15. We measure the runtime and costs of the various components of KGoT and illustrate them in Figure 17. We also provide insights into the tool usage, starting with the number of tasks for which a specific tools is used and whether that task was successful or not (see Figure 16). A more detailed analysis into the tool selection is provided in the plots of Figures 18 and 19 as well as the number of times the tools are used in Figure 20.

We provide now a brief explanation of the more opaque function names listed in Figure 17.

- **Any function marked as not logged** refers to function or tool calls that do not incur an LLM-related cost or where usage costs are logged within the tool itself.
- **WebSurfer.forward** submits a query to SerpApi.
- **Define Cypher query given new information** constructs a Cypher insert query based on newly gathered information.
- **Fix JSON** corrects malformed or invalid JSON for services like Neo4j.
- **Define forced retrieve queries** generates a Cypher retrieval query when the maximum number of iterations is reached.
- **Generate forced solution** generates a solution based on the state of the knowledge graph if no viable solution has been parsed after a Cypher retrieve or if the forced retrievals fails after exhausting all iterations.

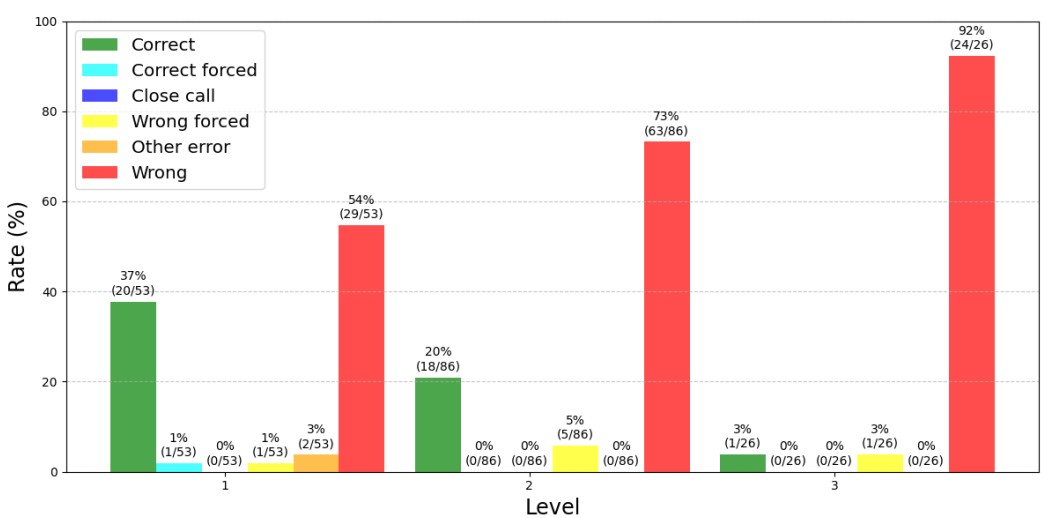

Figure 15: **Number of tasks per level that succeeded or fall into a given error category. Graph storage: Neo4j. Retrieval type: query. Model: GPT-4o mini.**

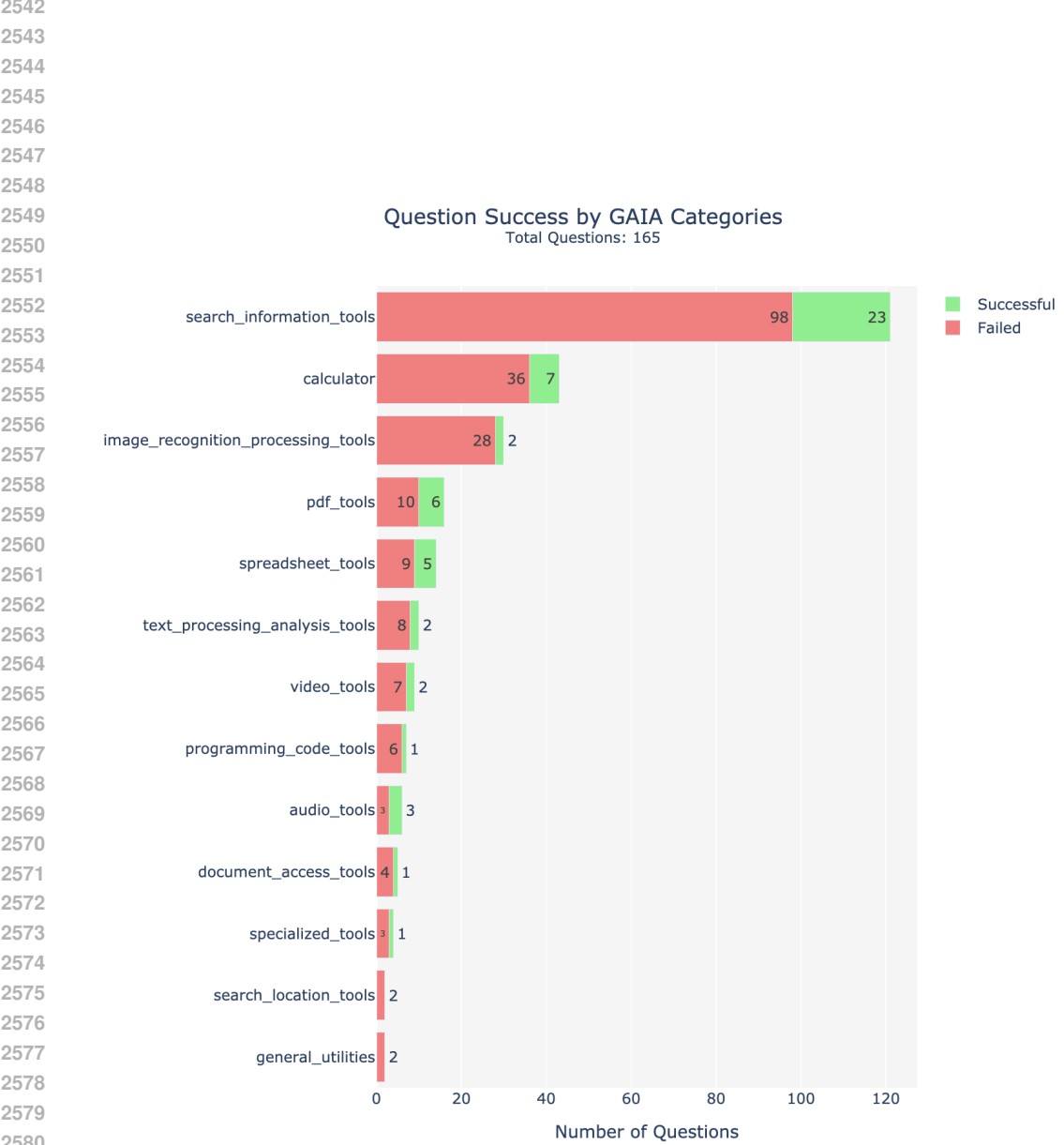

Figure 16: **Overview over how many tasks use a given tool and whether they are successful or not. Graph storage: Neo4j. Retrieval type: query. Model: GPT-4o mini.**

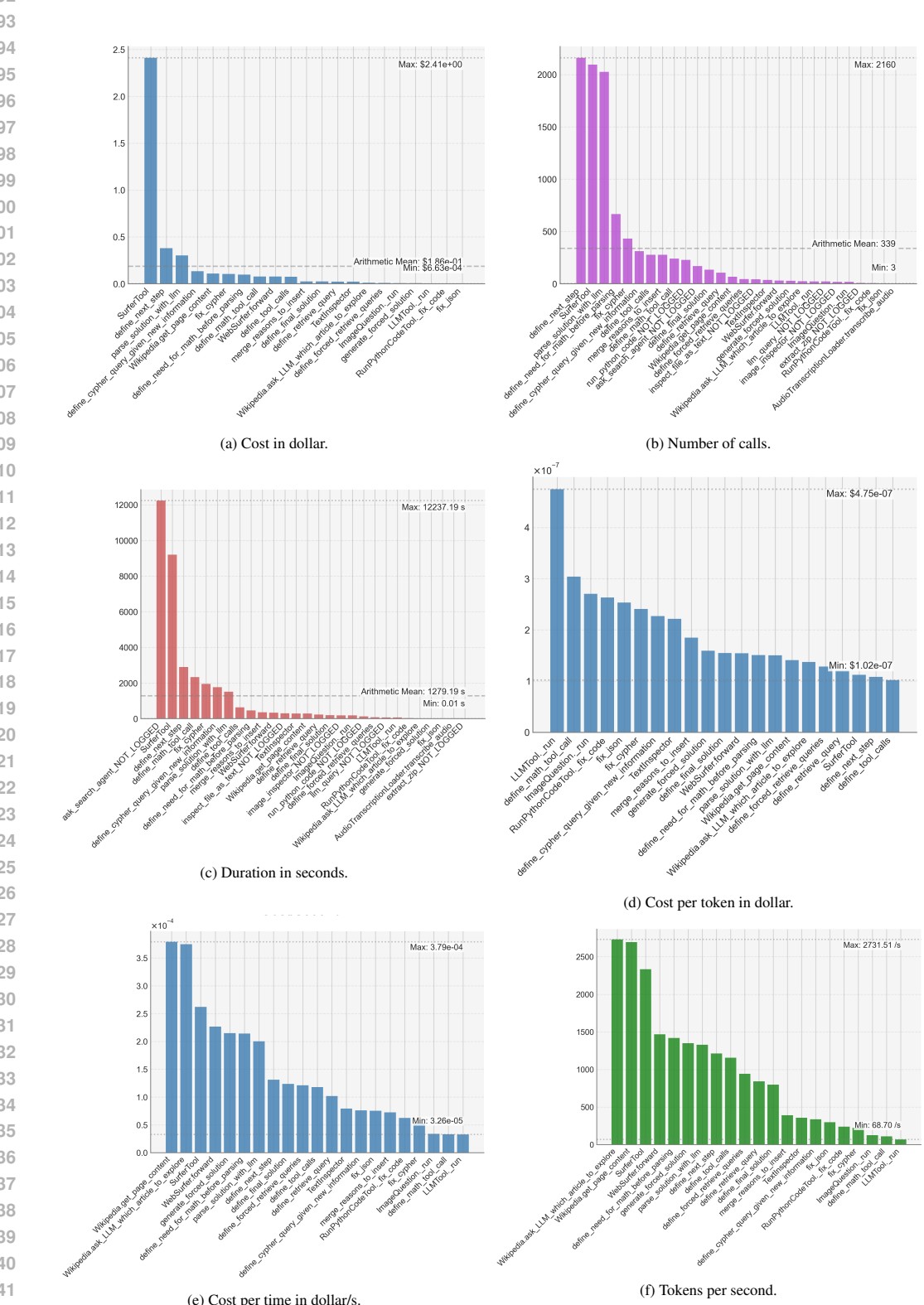

(a) Cost in dollar.

(b) Number of calls.

(c) Duration in seconds.

(d) Cost per token in dollar.

(e) Cost per time in dollar/s.

(f) Tokens per second.

Figure 17: **Overview over the execution time as well as the cost in dollar. Graph storage: Neo4j. Retrieval type: query. Model: GPT-4o mini.**

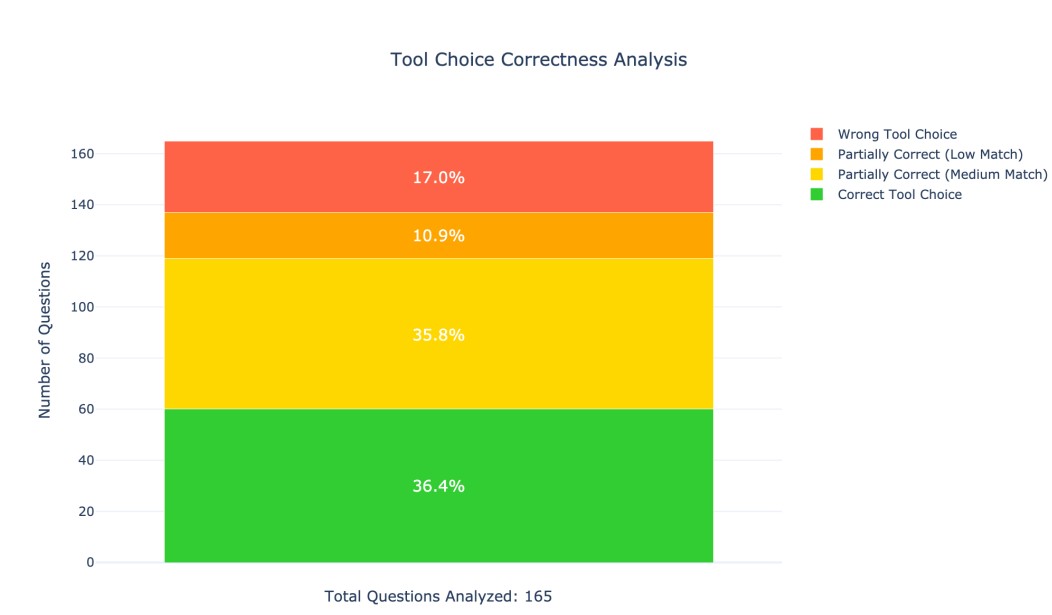

Figure 18: **Analysis of the tool selection. Graph storage: Neo4j. Retrieval type: query. Model: GPT-4o mini.**

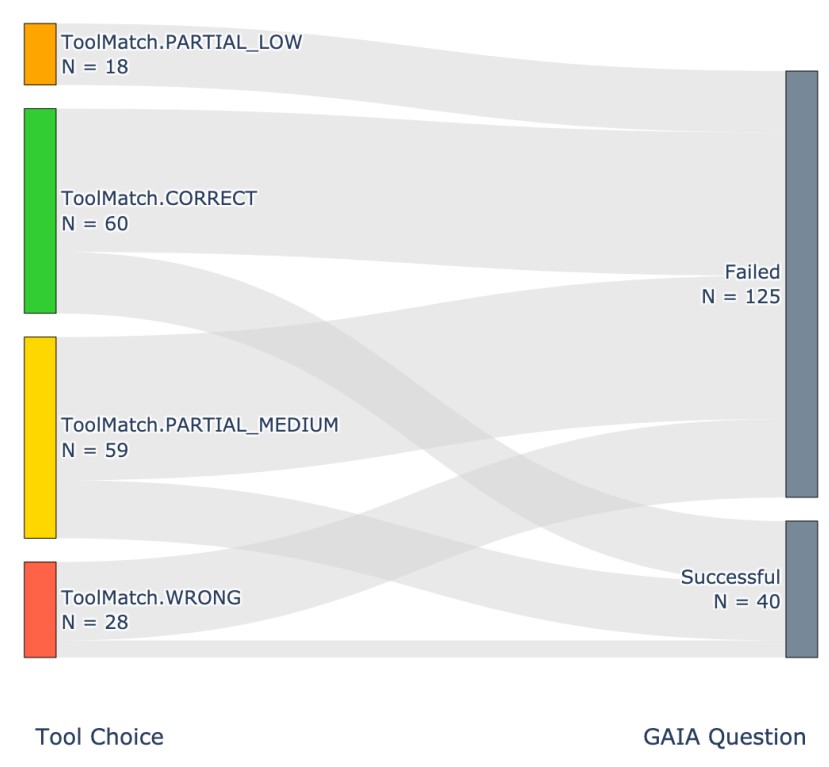

Figure 19: **Analysis of the tool selection. Graph storage: Neo4j. Retrieval type: query. Model: GPT-4o mini.**

Figure 20: **Analysis of the tool usage. Graph storage: Neo4j. Retrieval type: query. Model: GPT-4o mini.**

