# OpenReview forum: "Affordable AI Assistants with Knowledge Graph of Thoughts"
_ICLR.cc/2026/Conference — Submitted to ICLR 2026_

### Official Review · Reviewer_a4WT · 2025-10-30

**Soundness:** 3
**Presentation:** 2
**Contribution:** 2
**Rating:** 4
**Confidence:** 3

**Summary:**

This paper introduces Knowledge Graph of Thoughts (KGoT), an AI assistant architecture that integrates LLM reasoning with dynamically constructed knowledge graphs (KGs). The method aims to improve reasoning quality and reduce cost by transforming unstructured intermediate “thoughts” from LLMs into structured graph triples, which can then be queried via Cypher or SPARQL or embedded directly into context (“Direct Retrieval”).
Experiments show notable improvements over Hugging Face Agents and GPTSwarm on the GAIA benchmark (29% higher task success rate, ~36× cost reduction), and consistent gains on SimpleQA. The system is modular, incorporating asynchronous execution, MPI parallelization, and self-consistency mechanisms.

**Strengths:**

Interesting and well-motivated idea:
The concept of externalizing LLM reasoning into a structured Knowledge Graph of Thoughts is conceptually elegant and aligns with current directions in explainable and compositional reasoning.

Solid engineering effort:
The paper describes a comprehensive system integrating graph databases, asynchronous execution, and multi-tool orchestration (e.g., Neo4j, NetworkX, LangChain). The modular design and parallelism (Figure 2, 10) demonstrate non-trivial implementation maturity.

Empirical improvements:
Quantitative results (Figure 3, 5) show consistent improvements across GAIA and SimpleQA benchmarks in both accuracy and efficiency. The reported 36× cost reduction compared to GPT-4o is practically meaningful.

Transparency and reproducibility:
The paper includes detailed appendices with prompts, error-handling mechanisms, and architectural diagrams (Appendices B–C), suggesting reproducible research quality.

**Weaknesses:**

Lack of conceptual novelty beyond “Graph of Thoughts”:
The core idea—using knowledge graphs to structure intermediate reasoning—largely extends Graph of Thoughts (AAAI 2024) and similar prior works like Reasoning Language Models (Besta et al., 2025a). The paper reuses similar terminology (“thoughts” → “triples”), adding mainly an engineering layer around it. The contribution over these existing works is incremental rather than fundamentally new.

Limited analytical insight:
While the experiments are extensive, they remain empirical demonstrations without clear analysis of why the KG representation improves reasoning. For instance, Figure 5 shows gains when using KGs, but there is no ablation isolating specific design choices (e.g., triple quality, graph depth, or KG noise effects).

Evaluation may be superficial:
The GAIA benchmark mainly measures task success rather than true reasoning depth. The reported improvements (29%) might stem from better retrieval or tool orchestration, not necessarily improved reasoning capability. Further analysis (e.g., reasoning trace quality, error typology) would strengthen the claim.

Overly long and implementation-heavy:
A large portion of the paper (Sections B–C, ~20 pages) reads more like system documentation. While thorough, it detracts from the conceptual clarity and theoretical framing expected at ICLR. The main novelty could have been expressed much more concisely.

Unclear comparison fairness:
Some baselines (e.g., GraphRAG, GPTSwarm) are not re-implemented under identical tool sets, which makes the comparison potentially uneven. For example, KGoT integrates Hugging Face tools while also comparing against them.

**Questions:**

How does KGoT differ fundamentally from Graph of Thoughts (Besta et al., AAAI 2024)?

Can the authors provide a qualitative example where KG reasoning changes the causal reasoning path compared to vanilla chain-of-thought prompting?

What is the sensitivity of performance to graph size and triple accuracy?

How is the graph maintained across tasks—does it persist as a long-term memory or reset per query?

---

### Official Review · Reviewer_76Mr · 2025-10-31

**Soundness:** 2
**Presentation:** 2
**Contribution:** 3
**Rating:** 2
**Confidence:** 4

**Summary:**

The paper addresses the problem of high operational costs and low success rates in LLM-driven agents. The paper proposes an AI assistant architecture that reduces task execution costs while maintaining a high success rate. This approach integrates LLM reasoning with dynamically constructed KG. It allows the extraction and organization of task-relevant knowledge into a dynamic KG representation. Evaluation is conducted against top GAIA leaderboard baselines.

**Strengths:**

The problem addressed in the paper is the real world problem. The example presented in figure 1 is clear, the figures are well designed. The description of the evaluation and baseline are provided.

**Weaknesses:**

The current form of the paper has several issues which should be solved. Several elements are not defined to make the paper understandable. The paper claims their approach is an innovation, which is not the case.

Line 23: what is the meaning of 36 X?
Line 40-46: provide links to assistants. Provide examples of assistants
Line 70: Contribution 1 is not really an innovation because many works have addressed this as stated in the state of the art. The paper should position the contribution to these works instead of saying that this is an innovation.
Line 81-84: Yes, and that is also called RAG. It is not an innovation
Line 100-101: provide a reference

The figures should be positioned near their first mention to improve the readability of the paper.

Related work is not well presented. The paper lists many references, but does not make an effort to summarize, categorize, or compare them, which makes it difficult to understand the contribution of this work in relation to existing literature.

**Questions:**

How to decide if the system needs additional information or if information is already stored in the KG?
Line 64: How the graph construction cost at each query is evaluated?
Line 75-76: Yes, but what is the cost of the KG construction at every query? Which method is used for the KG construction?

---

### Official Review · Reviewer_Y8t7 · 2025-11-03

**Soundness:** 4
**Presentation:** 3
**Contribution:** 3
**Rating:** 6
**Confidence:** 3

**Summary:**

This paper proposed Knowledge Graph of Thought (KGoT), an agentic system design that integrates knowledge graph into the LLM agent's working memory, so as to save cost and increase effectiveness of LLM assistants. Both algorithmic design and system implementation is provided in the paper. Through evaluations on benchmarks including GAIA, the authors demonstrate that KGoT can improve task success rate while also reducing the total cost.

**Strengths:**

1. Comprehensive and technically detailed.
The paper provides both algorithmic and system-level contributions. The description of the KGoT framework covers reasoning logic, system architecture, and practical implementation details, supported by available codebases. This makes it a technically solid and reproducible piece of work.

2. Clear presentation and illustrative examples.
The manuscript is well organized, with clear writing and helpful figures that illustrate the key idea of transforming reasoning chains into structured knowledge graphs. The example from the GAIA benchmark (Figure 1) effectively clarifies the workflow.

3. Conceptually simple yet impactful idea.
The core concept—representing LLM reasoning as an evolving knowledge graph—is both intuitive and effective. It bridges structured symbolic reasoning with neural language models in an elegant way, making the method easy to extend.

4. Thorough empirical validation.
The experimental section is extensive, covering multiple baselines (HF Agents, GPTSwarm, Magentic-One, RAG variants) and benchmarks (GAIA, SimpleQA). The results show consistent improvements in both task success rate and cost efficiency.

**Weaknesses:**

1. Need for deeper analytical discussion.
The evaluation primarily focuses on benchmark-level outcomes (task count and cost). A more detailed breakdown of error cases or reasoning failures would strengthen the understanding of why KGoT succeeds or fails in certain task types.

2. Trade-off characterization.
While the paper emphasizes affordability, it lacks a quantitative comparison of latency and throughput against direct or zero-shot methods. It would be useful to show whether KGoT’s iterative graph construction significantly increases inference time relative to simpler agents.

**Questions:**

1. Can you further explain how the cost is saved by integrating KGoT? At which specific step does the cost reduction contribute the most?

---

### Official Review · Reviewer_CP4D · 2025-11-04

**Soundness:** 3
**Presentation:** 3
**Contribution:** 3
**Rating:** 4
**Confidence:** 5

**Summary:**

This paper proposes KGoT (Knowledge Graph of Thoughts), a framework that transforms unstructured task information into structured knowledge graphs, allowing smaller LLMs to reason and retrieve over them. The idea is intuitive and addresses a real cost bottleneck in AI assistants, showing impressive improvements on the GAIA benchmark in both success rate and efficiency.

**Strengths:**

The paper targets a real and pressing issue: current general-purpose AI assistants are costly and have limited success on GAIA; the work also provides a concrete cost backdrop (e.g., running the GAIA dev set with a frontier model is expensive). Unstructured sources (webpages, PDFs, etc.) are converted into a structured knowledge graph (KG), and a small model retrieves/reasons over the KG. The pipeline is modular and extensible to various graph query languages and backends. On GAIA, the approach reports higher success rates than HF Agents + GPT-4o-mini while being substantially cheaper (dozens of times lower), with end-to-end costs dropping by orders of magnitude in case studies. Comparisons include HF Agents, Magentic-One, GPTSwarm, two RAG baselines (Simple RAG and GraphRAG), and zero-shot settings, plus discussion on why GraphRAG underperforms on heterogeneous GAIA tasks.

**Weaknesses:**

1. In my reading, the primary failure mode lies in the graph-query stage. This design invites cascading errors from query construction to execution, and the proposed remedies seem cumbersome and complex. Given their centrality to end-to-end performance, these components should occupy a much larger portion of the paper—with clearer methodology and empirical validation.

2. For a graph-centric approach, it is important to include established KG QA benchmarks such as CWQ and WebQSP. Please add experiments on CWQ/WebQSP—comparing both performance and cost/latency—against graph-specific baselines and cost per correct answer. This would stress-test the query component under fixed schemas and validate the claimed cost-efficiency beyond GAIA.

**Questions:**

See weaknesses above.

---

### Official Review · Reviewer_7Gsj · 2025-11-07

**Soundness:** 2
**Presentation:** 3
**Contribution:** 2
**Rating:** 4
**Confidence:** 4

**Summary:**

This paper introduces Knowledge Graph of Thoughts (KGoT), a novel AI assistant framework designed to solve complex tasks efficiently with cost-effective LLMs. KGoT transforms unstructured problem-solving process into a dynamically constructed and evolving knowledge graph (KG) to represent the task state. The framework employs a dual LLM Controller (Graph Executor and Tool Executor) that iteratively enhance the KG by invoking external tools like web crawlers and code interpreters. Once KG is sufficient, the LLM solves the task by either directly embedding the KG into its context or generating a graph query to extract the answer. Experimental results on GAIA demonstrate that KGoT enables smaller models to solve more tasks while reducing API costs.

**Strengths:**

# Strength
- A simple and effective approach that transform unstructured task statements and intermediate reasoning steps into an evolving KG. This structured knowledge provides a mechanism to mitigate the state-tracking and multi-step reasoning failures in smaller model.
- The paper is well-organized with insightful visualizations that effectively illustrate the model performance
- The paper provides a thorough internal analysis comparing different knowledge extraction methods and backend, providing evidence for the strengths of these design choices.

**Weaknesses:**

# Weakness
- The paper primary comparison target **HF Agents**, is positioned as a key baseline. However, this is misleading. The cite source point to a personal GitHub repository, not an official hugging face agent framework. The author should compare KGoT to the official HF agent framework, **smolagents**[1], to demonstrate its superiority over the true SOTA in agent engineering.
- While the paper includes several agent-framework baselines and RAG variants, it omits comparisons with more recent methods [2]][3][4]
- The claim of affordable is a key selling point, but it’s based on total operation cost using a cheap model. This obscures the overhead cost of the KGoT framework itself. Majority voting, insight generation, tool calls, Cypher query generation, and solution parsing all seem to require numerous LLM calls per task. It would be useful to include the average token/cost breakdown that isolates framework overhead.
- The paper’s strongest results rely on fusion runs, which are described as “simulating the effect” of both backends. This suggests an ensemble result rather than a single deployable agent architecture, and author should provide more detail about these runs and include a computation time comparison.

[1] Roucher, A., del Moral, A. V., Wolf, T., von Werra, L., and Kaunismäki, E. ‘smolagents‘: a smol library to build great agentic systems. https://github.com/huggingface/smolagents, 2025.

[2] Tang, J., Fan, T., & Huang, C. (2025). AutoAgent: A Fully-Automated and Zero-Code Framework for LLM Agents. arXiv preprint arXiv:2502.05957.

[3] Hu, M., Zhou, Y., Fan, W., Nie, Y., Xia, B., Sun, T., ... & Li, G. (2025). Owl: Optimized workforce learning for general multi-agent assistance in real-world task automation. arXiv preprint arXiv:2505.23885.

[4] Zhu, H., Qin, T., Zhu, K., Huang, H., Guan, Y., Xia, J., ... & Zhou, W. Oagents: An empirical study of building effective agents, 2025.

**Questions:**

Please see weakness.

---

### Meta-Review · Area_Chair_VpAk · 2026-01-05

**Summary:**

There are some critical concerns raised by the reviewers, such as the need for deeper analytical discussion, bad structure, and insufficient organization of related works.

**Reviewer Concerns:**

The authors did not provide rebuttal.

**Reviewer Scores:**

They will not change their scores.

---

### Decision · Program_Chairs · 2026-01-26

Reject